# Learning Adversarial MDPs with Stochastic Hard Constraints

**Francesco Emanuele Stradi** [1]   **Matteo Castiglioni** [1]   **Alberto Marchesi** [1]   **Nicola Gatti** [1]

## Abstract

We study online learning in *constrained Markov decision processes* (CMDPs) with *adversarial losses* and *stochastic hard constraints*, under *bandit* feedback. We consider three scenarios. In the first one, we address general CMDPs, where we design an algorithm attaining sublinear regret and cumulative *positive constraints violation*. In the second scenario, under the mild assumption that a policy strictly satisfying the constraints exists and is known to the learner, we design an algorithm that achieves sublinear regret while ensuring that constraints are satisfied *at every episode* with high probability. In the last scenario, we only assume the *existence* of a strictly feasible policy, which is *not* known to the learner, and we design an algorithm attaining sublinear regret and *constant* cumulative positive constraints violation. Finally, we show that in the last two scenarios, a dependence on the Slater's parameter is unavoidable. To the best of our knowledge, our work is the first to study CMDPs involving both adversarial losses and hard constraints. Thus, our algorithms can deal with general non-stationary environments subject to requirements much stricter than those manageable with existing ones, enabling their adoption in a much wider range of applications.

## 1. Introduction

Reinforcement learning (Sutton & Barto, 2018) studies problems where a learner sequentially takes actions in an environment modeled as a *Markov decision process* (MDP) (Puterman, 2014). Most of the algorithms for such problems focus on learning policies that prescribe the learner how to take actions so as to minimize losses (equivalently, maximize rewards). However, in many real-world applications, the learner must fulfill additional requirements. For instance,

autonomous vehicles must avoid crashing (Wen et al., 2020; Isele et al., 2018), bidding agents in ad auctions must not deplete their budget (Wu et al., 2018; He et al., 2021), and recommendation systems must not present offending items to their users (Singh et al., 2020). A commonly-used model that allows to capture such additional requirements is the *constrained* MDP (CMDP) (Altman, 1999), where the goal is to learn a loss-minimizing policy while at the same time satisfying some constraints.

We study online learning problems in *episodic* CMDPs with *adversarial losses* and *stochastic hard constraints*, under *bandit* feedback. In such settings, the goal of the learner is to minimize their *regret*—the difference between their cumulative loss and what they would have obtained by always selecting a best-in-hindsight policy—, while at the same time guaranteeing that the constraints are satisfied during the learning process. We consider three scenarios that differ in the way in which constraints are satisfied and are all usually referred to as *hard* constraints settings in the literature (Liu et al., 2021; Guo et al., 2022). In the first scenario, the learner attains *sublinear* cumulative *positive* constraints violation. In the second one, the learner satisfies constraints at every episode, while, in the third one, they achieve *constant* cumulative *positive* constraints violation.

To the best of our knowledge, our work is the first to study CMDPs that involve both adversarial losses and hard constraints. Indeed, all the works on adversarial CMDPs (see, *e.g.*, (Wei et al., 2018; Qiu et al., 2020)) consider settings with *soft* constraints. These are much weaker than hard constraints, as they are only concerned with the minimization of the cumulative (both positive and negative) constraints violation. As a result, they allow negative violations to cancel out positive ones across different episodes. Such cancellations are unreasonable in real-world applications. For instance, in autonomous driving, avoiding a collision clearly does *not* "repair" a crash occurred previously. Furthermore, the only few works addressing stochastic hard constraints in CMDPs (Liu et al., 2021; Shi et al., 2023; Müller et al., 2024; Stradi et al., 2025) are restricted to *stochastic losses*. Thus, their techniques cannot be easily generalized to our setting. Our CMDP settings capture many more applications than theirs, since being able to deal with adversarial losses allows to tackle general non-stationary environments, which are ubiquitous in the real world.

---

[1]DEIB, Politecnico di Milano, Milan, Italy. Correspondence to: Francesco Emanuele Stradi <francescoemanuele.stradi@polimi.it>.

*Proceedings of the $42^{nd}$ International Conference on Machine Learning*, Vancouver, Canada. PMLR 267, 2025. Copyright 2025 by the author(s).

We refer to Appendix A for a complete discussion on related works.

### 1.1. Original contributions

We start by addressing the first scenario, where we design an algorithm—called Sublinear Violation Optimistic Policy Search (SV-OPS)—that guarantees both sublinear regret and sublinear cumulative positive constraints violation. SV-OPS builds on top of state-of-the-art learning algorithms in adversarial, unconstrained MDPs, by introducing the tools necessary to deal with constraints violation. Specifically, SV-OPS works by selecting policies that *optimistically* satisfy the constraints. SV-OPS updates the set of such policies in an online fashion, guaranteeing that it is always non-empty with high probability and that it collapses to the (true) set of constraints-satisfying policies as the number of episodes increases. This allows SV-OPS to attain sublinear violation. Crucially, even though such an "optimistic" set of policies changes during the execution of the algorithm, it always contains the (true) set of constraints-satisfying policies. This allows SV-OPS to attain sublinear regret. SV-OPS also addresses a problem left open by Qiu et al. (2020), *i.e.*, learning with *bandit* feedback in CMDPs with adversarial losses and stochastic constraints. Indeed, SV-OPS goes even further, as Qiu et al. (2020) were only concerned with soft constraints, while SV-OPS is capable of managing *positive* constraints violation.

Next, we switch the attention to the second scenario, where we design a *safe* algorithm, *i.e.*, one that satisfies the constraints at every episode. To achieve this, we need to assume that the learner has knowledge about a policy strictly satisfying the constraints. Indeed, this is necessary even in simple stochastic multi-armed bandit settings, as shown in (Bernasconi et al., 2022). This scenario begets considerable additional challenges compared to the first one, since assuring safety extremely limits exploration capabilities, rendering techniques for adversarial, unconstrained MDPs inapplicable. Nevertheless, we design an algorithm—called Safe Optimistic Policy Search (S-OPS)—that attains sublinear regret while being safe with high probability. S-OPS works by selecting, at each episode, a suitable randomization between the policy that SV-OPS would choose and the (known) policy strictly satisfying the constraints. Crucially, the probability defining the randomization employed by the algorithm is carefully chosen in order to *pessimistically* account for constraints satisfaction. This guarantees that a sufficient amount of exploration is performed.

Then, in the third scenario, we design an algorithm that attains *constant* cumulative positive constraints violation and sublinear regret, by simply assuming that a policy strictly satisfying the constraints exists, but it is *not* known to learner. Our algorithm—called Constant Violation Optimistic Policy

Search (CV-OPS)— estimates such a policy and its associated constraints violation in a constant number of episodes. This is done by employing two no-regret algorithms. The first one with the objective of minimizing violation, and the second one with the goal of selecting the most violated constraint. A stopping condition that depends on the guarantees of both the no-regret algorithms enforces that the number of episodes used to estimate the desired policy is sufficient, while still being constant. After that, CV-OPS runs S-OPS with the estimated policy, attaining the desired results.

Finally, we provide a lower bound showing that any algorithm attaining $o(\sqrt{T})$ violation cannot avoid a dependence on the Slater's parameter in the regret bound. We believe that this result may be of independent interest, since it is not only applicable to our second and third settings, but also to other settings where a larger violation is allowed.

## 2. Preliminaries

### 2.1. Constrained Markov decision processes

We study *episodic constrained* MDPs (CMDPs) (Altman, 1999) with *adversarial losses* and *stochastic constraints*. These are tuples $M := \left(X, A, P, \{\ell_t\}_{t=1}^T, \{G_t\}_{t=1}^T, \alpha\right)$. $T$ is the number of episodes.[1] $X$ and $A$ are finite state and action spaces. $P : X \times A \times X \to [0,1]$ is a transition function, where $P(x'|x, a)$ denotes the probability of moving from state $x \in X$ to $x' \in X$ by taking $a \in A$.[2] $\{\ell_t\}_{t=1}^T$ is the sequence of vectors of losses at each episode, namely $\ell_t \in [0,1]^{|X \times A|}$. We refer to the loss for a state-action pair $(x, a) \in X \times A$ as $\ell_t(x, a)$. Losses are adversarial, *i.e.*, no statistical assumption on how they are selected is made. $\{G_t\}_{t=1}^T$ is the sequence of matrices that define the *costs* characterizing the $m$ constraints at each episode, namely $G_t \in [0,1]^{|X \times A| \times m}$. For $i \in [m]$, the $i$-th constraint cost for $(x, a) \in X \times A$ is denoted by $g_{t,i}(x, a)$. Costs are stochastic, *i.e.*, the matrices $G_t$ are i.i.d. random variables distributed according to an (unknown) probability distribution $\mathcal{G}$. Finally, $\alpha = [\alpha_1, \ldots, \alpha_m] \in [0, L]^m$ is the vector of cost *thresholds* characterizing the $m$ constraints, where $\alpha_i$ denotes the threshold for the $i$-th constraint.

The learner uses a *policy* $\pi : X \times A \to [0,1]$, which defines a probability distribution over actions at each state. We denote by $\pi(\cdot|x)$ the distribution at $x \in X$, with $\pi(a|x)$ being

---

[1] We denote an episode by $t \in [T]$, where $[a \ldots b]$ is the set of all integers from $a$ to $b$ and $[b] := [1 \ldots b]$.

[2] For ease of notation, we focus w.l.o.g. on *loop-free* CMDPs. This means that $X$ is partitioned into $L + 1$ layers $X_0, \ldots, X_L$ with $X_0 = \{x_0\}$ and $X_L = \{x_L\}$. Moreover, the loop-free property requires that $P(x'|x, a) > 0$ only if $x' \in X_{k+1}$ and $x \in X_k$ for some $k \in [0 \ldots L - 1]$. Any (episodic) CMDP with horizon $H$ that is *not* loop-free can be cast into a loop-free one by duplicating the state space $H$ times. In loop-free CMDPs, we let $k(x) \in [0 \ldots L]$ be the layer index in which state $x \in X$ is.

---

**Algorithm 1** CMDP Interaction at episode $t \in [T]$

---

1: $\ell_t$, $G_t$ chosen *adversarially* and *stochastically*, resp.
2: Learner chooses a policy $\pi_t : X \times A \to [0, 1]$
3: Environment is initialized to state $x_0$
4: **for** $k = 0, \ldots, L - 1$ **do**
5:  Learner takes action $a_k \sim \pi_t(\cdot|x_k)$
6:  Learner sees $\ell_t(x_k, a_k), g_{t,i}(x_k, a_k) \forall i \in [m]$
7:  Environment evolves to $x_{k+1} \sim P(\cdot|x_k, a_k)$
8:  Learner observes the next state $x_{k+1}$
9: **end for**

---

the probability of action $a \in A$. Algorithm 1 details the learner-environment interaction at episode $t \in [T]$, where the learner has *bandit feedback*. Specifically, the learner observes the trajectory of state-action pairs $(x_k, a_k)$, for $k \in [0 \ldots L - 1]$, visited during the episode, their losses $\ell_t(x_k, a_k)$, and costs $g_{t,i}(x_k, a_k)$ for $i \in [m]$. The learner knows $X$ and $A$, but they do *not* know anything about the transition function $P$.

We introduce the notion of *occupancy measure* (Rosenberg & Mansour, 2019a). Given a transition function $P$ and a policy $\pi$, the vector $q^{P,\pi} \in [0,1]^{|X \times A \times X|}$ is the occupancy measure induced by $P$ and $\pi$. For every $x \in X_k, a \in A$, and $x' \in X_{k+1}$ with $k \in [0 \ldots L - 1]$, it holds $q^{P,\pi}(x, a, x') :=$ $\mathbb{P}[x_k = x, a_k = a, x_{k+1} = x'|P, \pi]$. Moreover, we also let $q^{P,\pi}(x, a) := \sum_{x' \in X_{k+1}} q^{P,\pi}(x, a, x')$ and $q^{P,\pi}(x) :=$ $\sum_{a \in A} q^{P,\pi}(x, a)$. Next, we define *valid* occupancies.

**Lemma 2.1** (Rosenberg & Mansour (2019b))**.** *A vector $q \in [0,1]^{|X \times A \times X|}$ is a valid occupancy measure of an episodic loop-free MDP if and only if it holds:*

$$
\begin{cases}
\sum_{x \in X_k} \sum_{a \in A} \sum_{x' \in X_{k+1}} q(x, a, x') = 1 \quad \forall k \in [0 \ldots L - 1] \\
\sum_{a \in A} \sum_{x' \in X_{k+1}} q(x, a, x') = \sum_{x' \in X_{k-1}} \sum_{a \in A} q(x', a, x) \\
\qquad\qquad\qquad\qquad\qquad \forall k \in [1 \ldots L - 1], \forall x \in X_k \\
P^q = P,
\end{cases}
$$

*where $P$ is the transition function of the MDP and $P^q$ is the one induced by $q$ (see below).*

Notice that any valid occupancy measure $q$ induces a transition function $P^q$ and a policy $\pi^q$, with $P^q(x'|x, a) =$ $q(x, a, x')/q(x, a)$ and $\pi^q(a|x) = q(x, a)/q(x)$.

### 2.2. Online learning with hard constraints

Our *baseline* for evaluating the performances of the learner is defined through a linear programming formulation of the (offline) learning problem in constrained MDPs. Specifically, given a constrained MDP $M := (X, A, P, \ell, G, \alpha)$ characterized by a loss vector $\ell \in [0, 1]^{|X \times A|}$, a cost matrix $G \in [0, 1]^{|X \times A| \times m}$, and a threshold vector $\alpha \in [0, L]^m$, such a problem consists in finding a policy minimizing

the loss while ensuring that all the constraints are satisfied. Thus, our baseline $\text{OPT}_{\ell,G,\alpha}$ is defined as the optimal value of a parametric linear program, which reads as follows:

$$
\text{OPT}_{\ell,G,\alpha} :=
\begin{cases}
\min_{q \in \Delta(M)} & \ell^\top q \quad \text{s.t.} \\
& G^\top q \leq \alpha,
\end{cases} \tag{1}
$$

where $q \in [0, 1]^{|X \times A|}$ is a vector encoding an occupancy measure, while $\Delta(M)$ is the set of valid occupancy measures. Notice that, given the equivalence between policy and occupancy, the (offline) learning problem can be formulated as a linear program working in the space of the occupancy measures $q$, since expected losses and costs are linear in $q$.

As customary in settings with adversarial losses, we measure the performance of an algorithm by comparing it with the *best-in-hindsight constraint-satisfying policy*. The performance of the learner is evaluated in terms of the *(cumulative) regret* $R_T := \sum_{t=1}^{T} \ell_t^\top q^{P,\pi_t} - T \cdot \text{OPT}_{\overline{\ell},\overline{G},\alpha}$, where $\overline{\ell} := \frac{1}{T} \sum_{t=1}^{T} \ell_t$ is the average of the adversarial losses over the $T$ episodes and $\overline{G} := \mathbb{E}_{G \sim \mathcal{G}}[G]$ is the expected value of the stochastic cost matrices. We let $q^*$ be a best-in-hindsight constraint-satisfying occupancy measure, *i.e.*, one achieving value $\text{OPT}_{\overline{\ell},\overline{G},\alpha}$, while we let $\pi^*$ be its corresponding policy. Thus, the regret reduces to $R_T := \sum_{t=1}^{T} \ell_t^\top (q^{P,\pi_t} - q^*)$. For the ease of notation, we refer to $q^{P,\pi_t}$ by simply using $q_t$, thus omitting the dependency on $P$ and $\pi_t$. Our goal is to design learning algorithms with $R_T = o(T)$, while at the same time satisfying constraints. We consider three different settings, all falling under the umbrella of *hard constraints* settings (Guo et al., 2022), introduced in the following.

#### 2.2.1. GUARANTEEING SUBLINEAR VIOLATION

In this setting, we consider the *(cumulative) positive constraints violation* $V_T := \max_{i \in [m]} \sum_{t=1}^{T} [\overline{G}^\top q_t - \alpha]_i^+$, where $[x]^+ := \max\{0, x\}$. Our goal is to design algorithms with $V_T = o(T)$. To achieve such a goal, we only need to assume that the problem is well posed, as follows:

**Assumption 2.2.** There is an occupancy measure $q^\diamond$, called *feasible solution*, such that $\overline{G}^\top q^\diamond \leq \alpha$.

#### 2.2.2. GUARANTEEING SAFETY

In this setting, our goal is to design algorithms ensuring that the following *safety property* is met:

**Definition 2.3** (Safe algorithm)**.** An algorithm is *safe* if and only if $\overline{G}^\top q_t \leq \alpha$ for all $t \in [T]$.

As shown by Bernasconi et al. (2022), without further assumptions, it is *not* possible to achieve $R_T = o(T)$ while at the same time guaranteeing that the safety property holds with high probability, even in simple stochastic multi-armed bandit instances. To design safe learning algorithms, we

need the following two assumptions. The first one is about the possibility of *strictly* satisfying constraints.

**Assumption 2.4** (Slater's condition)**.** There is an occupancy measure $q^\diamond : \overline{G}^\top q^\diamond < \alpha$. We call $q^\diamond$ *strictly feasible solution*, while a $\pi^\diamond$ induced by $q^\diamond$ is a *strictly feasible policy*.

The second assumption is related to learner's knowledge about a strictly feasible policy.

**Assumption 2.5.** Both the policy $\pi^\diamond$ and its costs $\beta = [\beta_1, \ldots, \beta_m] := \overline{G}^\top q^\diamond$ are known to the learner.

Intuitively, Assumption 2.5 is needed to guarantee that safety holds during the first episodes, when the leaner's uncertainty about the costs is high. Assumptions 2.4 and 2.5 are often employed in CMDPs (see, *e.g.*, (Liu et al., 2021)), as they are usually met in real-world applications of interest, where it is common to have access to a "do-nothing" policy resulting in *no* constraint being violated.

### 2.2.3. GUARANTEEING CONSTANT VIOLATION

In this setting, we relax Assumption 2.5, and we only assume Slater's condition (Assumption 2.4). We show that it is possible to achieve constant violation, namely $V_T$ is upper bounded by a constant independent of $T$, while attaining similar regret guarantees compared to the second setting. Our results depend on the Slater's parameter $\rho \in [0, L]$, which is defined as $\rho := \max_{q \in \Delta(M)} \min_{i \in [m]} \left[ \alpha_i - \bar{g}_i^\top q \right]$, with $q^\diamond := \arg\max_{q \in \Delta(M)} \min_{i \in [m]} \left[ \alpha_i - \bar{g}_i^\top q \right]$ being its associated occupancy. Given $\beta_i := \bar{g}_i^\top q^\diamond$, we have $\rho = \min_{i \in [m]} \left[ \alpha_i - \beta_i \right]$. By Assumption 2.4, it holds $\rho > 0$.

## 3. Concentration bounds

In this section, we provide concentration bounds for the estimates of unknown stochastic parameter of the CMDP. Let $N_t(x, a)$ be the number of episodes up to $t \in [T]$ in which the state-action pair $(x, a) \in X \times A$ is *visited*. Then, $\hat{g}_{t,i}(x, a) := \frac{\sum_{\tau \in [t]} g_{\tau,i}(x,a) \mathbb{1}_\tau \{x, a\}}{\max\{1, N_t(x,a)\}}$, with $\mathbb{1}_\tau \{x, a\} = 1$ if and only if $(x, a)$ is visited in episode $\tau$, is an unbiased estimator of the expected cost of constraint $i \in [m]$ for $(x, a)$, which we denote by $\bar{g}_i(x, a) := \mathbb{E}_{G \sim \mathcal{G}}[g_{t,i}(x, a)]$. Thus, by applying Hoeffding's inequality, it holds, with probability at least $1 - \delta$ that $|\hat{g}_{t,i}(x, a) - \bar{g}_i(x, a)| \le \xi_t(x, a)$, where we let the confidence bound $\xi_t(x, a) := \min\{1, \sqrt{4 \ln(T|X||A|m/\delta)/\max\{1, N_t(x, a)\}}\}$ (refer to Lemma B.2 for the formal result). For ease of notation, we let $\hat{G}_t \in [0, 1]^{|X \times A| \times m}$ be the matrix of the estimated costs $\hat{g}_{t,i}(x, a)$. Moreover, we denote by $\xi_t \in [0, 1]^{|X \times A|}$ the vector whose entries are the bounds $\xi_t(x, a)$, and we let $\Xi_t \in [0, 1]^{|X \times A| \times m}$ be a matrix built by concatenating vectors $\xi_t$ in such a way that the statement of Lemma B.2 becomes: $|\hat{G}_t - \overline{G}| \preceq \Xi_t$ holds with probability at least $1 - \delta$, where $|\cdot|$ and $\preceq$ are applied component wise. In

the following, given any $\delta \in (0, 1)$, we refer to the event defined in Lemma B.2 as $\mathcal{E}^G(\delta)$.

Similarly, we define *confidence sets* for the transition function of a CMDP, by exploiting suitable concentration bounds for estimated transition probabilities. By letting $M_t(x, a, x')$ be the total number of episodes up to $t \in [T]$ in which $(x, a) \in X \times A$ is visited and the environment transitions to $x' \in X$, the estimated transition probability for $(x, a, x')$ is defined as $\hat{P}_t(x' \mid x, a) := \frac{M_t(x,a,x')}{\max\{1, N_t(x,a)\}}$. Then, at episode $t \in [T]$, the confidence set for the transitions is $\mathcal{P}_t := \bigcap_{(x,a,x') \in X \times A \times X} \mathcal{P}_t^{x,a,x'}$, with $\mathcal{P}_t^{x,a,x'} := \left\{ \overline{P} : |\overline{P}(x'|x,a) - \hat{P}_t(x'|x,a)| \le \epsilon_t(x, a, x') \right\}$, where we let $\epsilon_t(x, a, x') := 2\sqrt{\frac{\hat{P}_t(x'|x,a) \ln(T|X||A|/\delta)}{\max\{1, N_t(x,a)-1\}}} + \frac{14 \ln(T|X||A|/\delta)}{3 \max\{1, N_t(x,a)-1\}}$ for some confidence $\delta \in (0, 1)$. It is well known that, with probability at least $1 - 4\delta$, it holds that the transition function $P$ belongs to $\mathcal{P}_t$ for all $t \in [T]$ (see (Jin et al., 2020) and Lemma G.1 for the formal statement). At each $t \in [T]$, given a confidence set $\mathcal{P}_t$, it is possible to efficiently build a set $\Delta(\mathcal{P}_t)$ that comprises all the occupancy measures that are valid with respect to every transition function $\overline{P} \in \mathcal{P}_t$. For reasons of space, we defer the formal definition of $\Delta(\mathcal{P}_t)$ to Appendix G. Lemma G.1 implies that, with high probability, the set $\Delta(M)$ of valid occupancy measures is included in all the "estimated" sets $\Delta(\mathcal{P}_t)$, for every $t \in [T]$. In the following, given any confidence $\delta \in (0, 1)$, we refer to the event that $\Delta(M) \subseteq \bigcap_{t \in [T]} \Delta(\mathcal{P}_t)$ as $\mathcal{E}^\Delta(\delta)$, which holds with probability at least $1 - 4\delta$ thanks to Lemma G.1.

Finally, for ease of presentation, given $\delta \in (0, 1)$ we define a *clean event* $\mathcal{E}^{G,\Delta}(\delta)$ under which all the concentration bounds for costs and transitions correctly hold. Formally, $\mathcal{E}^{G,\Delta}(\delta) := \mathcal{E}^G(\delta) \cap \mathcal{E}^\Delta(\delta)$, which holds with probability at least $1 - 5\delta$ by a union bound (and Lemmas B.2 and G.1).

## 4. Guaranteeing sublinear violation

We start by designing the SV-OPS algorithm, guaranteeing that both the regret $R_T$ and the *positive* constraints violation $V_T$ are sublinear in $T$. To get this result, we only need the existence of a feasible solution (Assumption 2.2). Dealing with adversarial losses while limiting positive constraints violation begets considerable challenges, which go beyond classical exploration-exploitation trade-offs faced in unconstrained settings. On the one hand, using state-of-the-art algorithms for online learning in adversarial, unconstrained MDPs would lead to sublinear regret, but violation would grow linearly. On the other hand, a naïve approach that randomly explores to compute a set of policies satisfying the constraints with high probability can lead to sublinear violation, at the cost of linear regret. Thus, a clever adaptation of the techniques for unconstrained settings is needed.

**Algorithm 2** `SV-OPS`

---

**Require:** $X, A, \alpha, T, \delta, \eta, \gamma$
1: **for** $k \in [0 \ldots L-1]$, $(x, a, x') \in X_k \times A \times X_{k+1}$ **do**
2:   $N_0(x, a) \leftarrow 0$; $M_0(x, a, x') \leftarrow 0$
3:   $\widehat{q}_1(x, a, x') \leftarrow \frac{1}{|X_k||A||X_{k+1}|}$
4: **end for**
5: $\pi_1 \leftarrow \pi^{\widehat{q}_1}$
6: **for** $t \in [T]$ **do**
7:   Choose $\pi_t$ in Algorithm 1 and receive feedback
8:   Build *upper occupancy bounds* for $k \in [0 \ldots L-1]$:

$$u_t(x_k, a_k) \leftarrow \max_{\overline{P} \in \mathcal{P}_{t-1}} q^{\overline{P}, \pi_t}(x_k, a_k)$$

9:   Build *optimistic loss estimator* for $(x, a) \in X \times A$:

$$\widehat{\ell}_t(x, a) \leftarrow \begin{cases} \frac{\ell_t(x,a)}{u_t(x,a)+\gamma} & \text{if } \mathbb{1}_t\{x, a\} = 1 \\ 0 & \text{otherwise} \end{cases}$$

10:   **for** $k \in [0 \ldots L-1]$ **do**
11:     $N_t(x_k, a_k) \leftarrow N_{t-1}(x_k, a_k) + 1$
12:     $M_t(x_k, a_k, x_{k+1}) \leftarrow M_{t-1}(x_k, a_k, x_{k+1}) + 1$
13:   **end for**
14:   Build $\mathcal{P}_t, \widehat{G}_t$, and $\Xi_t$ as in Section 3
15:   Build *unconstrained occupancy* for all $(x, a, x')$:

$$\tilde{q}_{t+1}(x, a, x') \leftarrow \widehat{q}_t(x, a, x') e^{-\eta \widehat{\ell}_t(x,a)}$$

16:   **if** $\text{PROJ}(\tilde{q}_{t+1}, \widehat{G}_t, \Xi_t, \mathcal{P}_t)$ is *feasible* **then**
17:     $\widehat{q}_{t+1} \leftarrow \text{PROJ}(\tilde{q}_{t+1}, \widehat{G}_t, \Xi_t, \mathcal{P}_t)$
18:   **else**
19:     $\widehat{q}_{t+1} \leftarrow$ any $q \in \Delta(\mathcal{P}_t)$
20:   **end if**
21:   $\pi_{t+1} \leftarrow \pi^{\widehat{q}_{t+1}}$
22: **end for**

---

Our algorithm—called Sublinear Violation Optimistic Policy Search (`SV-OPS`)—works by selecting policies derived from a set of occupancy measures that *optimistically* satisfy cost constraints. This ensures that the set is always non-empty with high probability and that it collapses to the (true) set of constraint-satisfying occupancy measures as the number of episodes increases, enabling `SV-OPS` to attain sublinear constraints violation. The fundamental property preserved by `SV-OPS` is that, even though the "optimistic" set changes during the execution of the algorithm, it always subsumes the (true) set of constraint-satisfying occupancy measures. This crucially allows `SV-OPS` to employ classical policy-selection methods for unconstrained MDPs.

Algorithm 2 provides the pseudocode of `SV-OPS`. At each episode $t \in [T]$, `SV-OPS` plays policy $\pi_t$ and receives feedback as described in Algorithm 1 (Line 7). Then, `SV-OPS` computes an *upper occupancy bound* $u_t(x_k, a_k)$ for every state-action pair $(x_k, a_k)$ visited during Algorithm 1, by using the confidence set for the transition function $\mathcal{P}_{t-1}$ computed in the previous episode, namely, it sets $u_t(x_k, a_k) :=$ $\max_{\overline{P} \in \mathcal{P}_{t-1}} q^{\overline{P}, \pi_t}(x, a)$ for every $k \in [0 \ldots L-1]$ (Line 8).

Intuitively, $u_t(x_k, a_k)$ represents the maximum probability with which $(x_k, a_k)$ is visited when using policy $\pi_t$, given the confidence set for the transition function built so far. The upper occupancy bounds are combined with the exploration factor $\gamma$ to compute an *optimistic loss estimator* $\widehat{\ell}_t(x, a)$ for every state-action pair $(x, a) \in X \times A$ (see Line 9). After that, `SV-OPS` updates all the counters given the path traversed in Algorithm 1 (Lines 11–12), it builds the new confidence set $\mathcal{P}_t$, and it computes the matrices $\widehat{G}_t$ and $\Xi_t$ of estimated costs and their bounds, respectively, by using the feedback (Line 14). To choose a policy $\pi_{t+1}$, `SV-OPS` first computes an *unconstrained occupancy measure* $\tilde{q}_{t+1}$ according to an unconstrained OMD update (Orabona, 2019) (see Line 15). Then, $\tilde{q}_{t+1}$ is projected onto a suitably-defined set of occupancy measures that *optimistically* satisfy the constraints. Next, we formally define the projection (Line 16).

$$\text{PROJ}(\tilde{q}_{t+1}, \widehat{G}_t, \Xi_t, \mathcal{P}_t) := \begin{cases} \underset{q \in \Delta(\mathcal{P}_t)}{\arg\min} & D(q||\tilde{q}_{t+1}) \\ \text{s.t.} & (\widehat{G}_t - \Xi_t)^\top q \leq \alpha, \end{cases} \quad (2)$$

where $D(q||\tilde{q}_{t+1})$ is the unnormalized KL-divergence between $q$ and $\tilde{q}_{t+1}$. Problem (2) is a linearly-constrained convex mathematical program, and, thus, it can be solved efficiently, that is, in polynomial time, for an arbitrarily-good approximate solution.[3] Intuitively, Problem (2) performs a projection onto the set of $q \in \Delta(\mathcal{P}_t)$ that additionally satisfy $(\widehat{G}_t - \Xi_t)^\top q \leq \alpha$, where lower confidence bounds $\widehat{G}_t - \Xi_t$ for the costs are used in order to take an optimistic approach with respect to constraints satisfaction. Finally, if Problem (2) is feasible, then at the next episode `SV-OPS` selects the $\pi^{\widehat{q}_{t+1}}$ induced by a solution $\widehat{q}_{t+1}$ to Problem (2) (Line 17), otherwise it chooses a policy induced by any $q \in \Delta(\mathcal{P}_t)$ (Line 19).

The optimistic approach adopted in Problem (2) crucially allows to prove the following lemma.

**Lemma 4.1.** *Given confidence $\delta \in (0, 1)$, Algorithm 2 ensures that $\text{PROJ}(\tilde{q}_{t+1}, \widehat{G}_t, \Xi_t, \mathcal{P}_t)$ is feasible at every episode $t \in [T]$ with probability at least $1 - 5\delta$.*

Lemma 4.1 holds since, under the event $\mathcal{E}^{G,\Delta}(\delta)$, projection is performed on a set subsuming the (true) set of constraints-satisfying occupancies. Lemma 4.1 is fundamental, as it allows to show that `SV-OPS` attains sublinear $V_T$ and $R_T$.

### 4.1. Cumulative positive constraints violation

To prove that the positive constraints violation achieved by `SV-OPS` is sublinear, we exploit the fact that the concentration bounds for costs and transitions shrink at a rate of $\mathcal{O}(1/\sqrt{T})$. This allows us to show the following result.

---

[3]As customary in adversarial MDPs, we assume that an optimal solution to Problem (2) can be computed efficiently. Otherwise, we can still derive all of our results up to small approximations.

**Theorem 4.2.** *Given $\delta \in (0,1)$, Algorithm 2 attains $V_T \leq$ $\mathcal{O}\left(L|X|\sqrt{|A|T \ln\left(T|X||A|m/\delta\right)}\right)$ with prob. at least $1 - 8\delta$.*

## 4.2. Cumulative regret

The crucial observation that allows us to prove that the regret attained by `SV-OPS` grows sublinearly is that the set on which the algorithm perform its projection step (Problem (2)) always contains the (true) set of occupancy measures that satisfy the constraints, and, thus, it also always contains the best-in-hindsight constraint-satisfying occupancy measure $q^*$. As a result, even though cost estimates may be arbitrarily bad, `SV-OPS` is still guaranteed to select policies resulting in losses that are smaller than or equal to those incurred by $q^*$. This allows us to show the following:

**Theorem 4.3.** *Given $\delta \in (0,1)$, by setting $\eta = \gamma = \sqrt{L \ln(L|X||A|/\delta)/T|X||A|}$, Algorithm 2 attains $R_T \leq$ $\mathcal{O}\left(L|X|\sqrt{|A|T \ln\left(T|X||A|/\delta\right)}\right)$ with prob. at least $1 - 10\delta$.*

## 5. Guaranteeing safety

We design another algorithm, called `S-OPS`, attaining sublinear regret and enjoying the safety property with high probability. To do this, we work under Assumptions 2.4 and 2.5. Designing safe algorithms raises many additional challenges compared to the case studied in Section 4. Indeed, adapting techniques for adversarial, unconstrained MDPs does *not* work anymore, and, thus, *ad hoc* approaches are needed. This is because safety extremely limits exploration.

Our algorithm—Safe Optimistic Policy Search (`S-OPS`)—builds on top of the `SV-OPS` algorithm developed in Section 4. Selecting policies derived from the "optimistic" set of occupancy measures, as done by `SV-OPS`, is *not* sufficient anymore, as it would clearly result in the safety property being unsatisfied during the first episodes. Our new algorithm circumvents such an issue by employing, at each episode, a suitable randomization between the policy derived from the "optimistic" set (the one `SV-OPS` would select) and the strictly feasible policy $\pi^\diamond$. Crucially, as we show next, such a randomization accounts for constraints satisfaction by taking a *pessimistic* approach, namely, by considering upper confidence bounds on the costs characterizing the constraints. This is needed in order to guarantee the safety property. Moreover, having access to the strictly feasible policy $\pi^\diamond$ and its expected costs $\beta$ (Assumption 2.5) allows `S-OPS` to always place a sufficiently large probability on the policy derived from the "optimistic" set, so that a sufficient amount of exploration is guaranteed, and, in its turn, sublinear regret is attained. Notice that `S-OPS` effectively selects *non-Markovian* policies, as it employs a randomization between two Markovian policies at each episode.

Algorithm 3 provides the pseudocode of `S-OPS`. Differently

---

**Algorithm 3** `S-OPS`

**Require:** $X, A, \alpha, T, \delta, \eta, \gamma, \pi^\diamond, \beta$
1: **for** $k \in [0 \ldots L-1], (x, a, x') \in X_k \times A \times X_{k+1}$ **do**
2:     $N_0(x,a) \leftarrow 0; M_0(x,a,x') \leftarrow 0$
3:     $\widehat{q}_1(x,a,x') \leftarrow \frac{1}{|X_k||A||X_{k+1}|}$
4: **end for**
5: $\pi_1 \leftarrow \begin{cases} \pi^\diamond & \text{w. probability } \lambda_0 := \max_{i \in [m]}\left\{\frac{L-\alpha_i}{L-\beta_i}\right\} \\ \pi^{\widehat{q}_1} & \text{w. probability } 1 - \lambda_0 \end{cases}$
6: **for** $t \in [T]$ **do**
7:     Select $\pi_t$ in Algorithm 1 and receive feedback
8:     Build *upper occupancy bounds* for $k \in [0 \ldots L-1]$:

$$u_t(x_k, a_k) \leftarrow \max_{\overline{P} \in \mathcal{P}_{t-1}} q^{\overline{P}, \pi_t}(x_k, a_k)$$

9:     Build *optimistic loss estimator* for $(x,a) \in X \times A$:

$$\widehat{\ell}_t(x,a) \leftarrow \begin{cases} \frac{\ell_t(x,a)}{u_t(x,a)+\gamma} & \text{if } \mathbb{1}_t\{x,a\} = 1 \\ 0 & \text{otherwise} \end{cases}$$

10:     **for** $k \in [0 \ldots L-1]$ **do**
11:       $N_t(x_k, a_k) \leftarrow N_{t-1}(x_k, a_k) + 1$
12:       $M_t(x_k, a_k, x_{k+1}) \leftarrow M_{t-1}(x_k, a_k, x_{k+1}) + 1$
13:     **end for**
14:     Build $\mathcal{P}_t, \widehat{G}_t$, and $\Xi_t$ as in Section 3
15:     Build *unconstrained occupancy* for all $(x,a,x')$:

$$\widetilde{q}_{t+1}(x,a,x') \leftarrow \widehat{q}_t(x,a,x')e^{-\eta\widehat{\ell}_t(x,a)}$$

16:     **if** $\text{PROJ}\left(\widetilde{q}_{t+1}, \widehat{G}_t, \Xi_t, \mathcal{P}_t\right)$ is *feasible* **then**
17:       $\widehat{q}_{t+1} \leftarrow \text{PROJ}\left(\widetilde{q}_{t+1}, \widehat{G}_t, \Xi_t, \mathcal{P}_t\right)$
18:       $\widehat{\pi}_{t+1} \leftarrow \pi^{\widehat{q}_{t+1}}$
19:       Build $\widehat{u}_{t+1} \in [0,1]^{|X \times A|}$ so that for all $(x,a)$:

$$\widehat{u}_{t+1}(x,a) \leftarrow \max_{\overline{P} \in \mathcal{P}_t} q^{\overline{P}, \widehat{\pi}_{t+1}}(x,a)$$

20:       $\sigma \leftarrow \max_{i \in [m]}\left\{\frac{\min\left\{(\widehat{g}_{t,i}+\xi_t)^\top \widehat{u}_{t+1}, L\right\}-\alpha_i}{\min\left\{(\widehat{g}_{t,i}+\xi_t)^\top \widehat{u}_{t+1}, L\right\}-\beta_i}\right\}$
21:       $\lambda_t \leftarrow \begin{cases} \sigma & \text{if } \exists i \in [m] : (\widehat{g}_{t,i}+\xi_t)^\top \widehat{u}_{t+1} > \alpha_i \\ 0 & \text{if } \forall i \in [m] : (\widehat{g}_{t,i}+\xi_t)^\top \widehat{u}_{t+1} \leq \alpha_i \end{cases}$
22:     **else**
23:       $\widehat{q}_{t+1} \leftarrow$ take any $q \in \Delta(\mathcal{P}_t); \lambda_t \leftarrow 1$
24:     **end if**
25:     $\pi_{t+1} \leftarrow \begin{cases} \pi^\diamond & \text{with probability } \lambda_t \\ \pi^{\widehat{q}_{t+1}} & \text{with probability } 1 - \lambda_t \end{cases}$
26: **end for**

---

from `SV-OPS`, the policy selected at the first episode is obtained by randomizing a uniform occupancy measure with $\pi^\diamond$ (Line 5). The probability $\lambda_0$ of selecting $\pi^\diamond$ is chosen pessimistically. Intuitively, in the first episode, being pessimistic means that $\lambda_0$ must guarantee that the constraints are satisfied for any possible choice of costs and transitions, and, thus, $\lambda_0 := \max_{i \in [m]}\left\{L-\alpha_i/L-\beta_i\right\}$. Thanks to Assumptions 2.4 and 2.5, it is always the case that $\lambda_0 < 1$. Thus, $\pi_1 \neq \pi^\diamond$ with positive probability and some explo-

ration is performed even in the first episode. Analogously to SV-OPS, at each $t \in [T]$, S-OPS selects a policy $\pi_t$ and receives feedback as described in Algorithm 1, it computes optimistic loss estimators, it updates the confidence set for the transitions, and it computes the matrices of estimated costs and their bounds. Then, as in SV-OPS, an update step of unconstrained OMD is performed. Although identical to the update done in SV-OPS, the one in S-OPS uses loss estimators computed when using a randomization between the policy obtained by solving Problem (2) and the strictly feasible policy $\pi^\diamond$. Thus, there is a mismatch between the occupancy measure used to estimate losses and the one computed by the projection step. The projection step performed by S-OPS (Line 16) is the same as the one in SV-OPS. Specifically, the algorithm projects the unconstrained occupancy measure $\tilde{q}_{t+1}$ onto an "optimistic" set by solving Problem (2), which, if the problem is feasible, results in occupancy measure $\hat{q}_{t+1}$. However, differently from SV-OPS, when the problem is feasible, S-OPS does *not* select the policy $\pi^{\hat{q}_{t+1}}$ derived from $\hat{q}_{t+1}$, but it rather uses a randomization between such a policy and the strictly feasible policy $\pi^\diamond$ (Line 25). The probability $\lambda_t$ of selecting $\pi^\diamond$ is chosen pessimistically with respect to constraints satisfaction, by using upper confidence bounds for the costs and upper occupancy bounds given policy $\pi^{\hat{q}_{t+1}}$ (Lines 19 and 21). Such a pessimistic approach ensures that the constraints are satisfied with high probability, thus making the algorithm safe with high probability. If Problem (2) is *not* feasible, then any occupancy measure in $\Delta(\mathcal{P}_t)$ can be selected (Line 23).

### 5.1. Safety property

We show that S-OPS is safe with high probability.

**Theorem 5.1.** *Given a confidence $\delta \in (0,1)$, Algorithm 3 is safe with probability at least $1 - 5\delta$.*

Intuitively, Theorem 5.1 follows from the way in which the randomization probability $\lambda_t$ is defined. Indeed, $\lambda_t$ relies on two crucial components: (i) a pessimistic estimate of the costs for state-action pairs, namely, the upper confidence bounds $\hat{g}_{t,i} + \xi_t$, and (ii) a pessimistic choice of transition probabilities, encoded by the upper occupancy bounds defined by the vector $\hat{u}_t$. Notice that the $\max_{i \in [m]}$ operator allows to be conservative with respect to all the constraints.

### 5.2. Cumulative regret

Proving that S-OPS attains sublinear regret begets challenges that, to the best of our knowledge, have never been addressed before. Specifically, analyzing the estimates of the adversarial losses requires non-standard techniques in our setting, since the policy $\pi_t$ used by the algorithm and determining the feedback is *not* the one resulting from an OMD-like update, as it is obtained via a non-standard randomization. Nevertheless, the particular shape of the prob-

---

**Algorithm 4** CV-OPS

**Require:** Anytime adversarial MDPs regret minimizer $\mathcal{A}^P$, online linear optimizer $\mathcal{A}^D$
1: **for** $t \in [T]$ **do**
2:      Select $\pi_t \leftarrow \mathcal{A}^P$
3:      Select $\phi_t \leftarrow \mathcal{A}^D$
4:      Play $\pi_t$ and observe feedback as prescribed in Algorithm 1
5:      Feed $\{x_k, a_k, \sum_{i \in [m]} \phi_{t,i}(g_{t,i}(x_k, a_k) - \frac{\alpha_i}{L})\}_{k=1}^{L-1}$ to $\mathcal{A}^P$
6:      Feed $\{-\sum_{k=1}^{L-1}(g_{t,i}(x_k, a_k) - \frac{\alpha_i}{L})\}_{i \in [m]}$ to $\mathcal{A}^D$
7:      **if** $-\max_{i \in [m]} \sum_{\tau \in [t]} \sum_{k=1}^{L-1}(g_{t,i}(x_k, a_k) - \frac{\alpha_i}{L}) \geq 2C_{\mathcal{A}}^P \sqrt{t \ln(t)} + 8L\sqrt{2t \ln \frac{1}{\delta}} + 2C_{\mathcal{A}}^D \sqrt{t}$ **then**
8:          Go to Line 11
9:      **end if**
10: **end for**

11: $\hat{\rho} \leftarrow -\frac{1}{t} \max_{i \in [m]} \sum_{\tau \in [t]} \left( \sum_{k=1}^{L-1} g_{t,i}(x_k, a_k) - \alpha_i \right) - \frac{2L}{t} \sqrt{2t \ln 1/\delta}$
12: $\hat{\pi}^\diamond \leftarrow \pi_\tau$ with probability $1/t$, for $\tau \in [t]$
13: **Run** S-OPS with $\beta_i = \alpha_i - \hat{\rho}$ for all $i \in [m]$ and $\pi^\diamond = \hat{\pi}^\diamond$

---

ability $\lambda_t$ can be exploited to overcome such a challenge. Indeed, we show that each $\lambda_t$ can be upper bounded by the initial $\lambda_0$, and, thus, a loss estimator from feedback received by using a policy computed by an OMD-like update is available with probability at least $1 - \lambda_0$. This observation is crucial in order ot prove the following result:

**Theorem 5.2.** *Given $\delta \in (0,1)$, by setting $\eta = \gamma = \sqrt{L \ln(L|X||A|/\delta)/T|X||A|}$, Algorithm 3 attains $R_T \leq \mathcal{O}\left( \Psi L^3 |X| \sqrt{|A|T \ln\left(T|X||A|m/\delta\right)} \right)$ with prob. at least $1 - 11\delta$, where $\Psi := \max_{i \in [m]}\{1/\min\{(\alpha_i - \beta_i), (\alpha_i - \beta_i)^2\}\}$.*

The regret bound in Theorem 5.2 is in line with the one of SV-OPS in the bounded violation setting, with an additional $\Psi L^2$ factor. Such a factor comes from the mismatch between loss estimators and the occupancy measure chosen by the OMD-like update. Notice that $\Psi$ depends on the violation gap $\min_{i \in [m]}\{\alpha_i - \beta_i\}$, which represents how much the strictly feasible solution satisfies the constraints. Such a dependence is expected, since the better the strictly feasible solution (in terms of constraints satisfaction), the larger the exploration performed during the first episodes.

## 6. Guaranteeing constant violation

In this section, we provide an algorithm that attains *constant* cumulative positive violation. To achieve this goal, we only need that a strictly feasible policy exists (Assumption 2.4). Algorithm 4 provides the pseudocode of Constant Violation Optimistic Policy Search (CV-OPS). The key idea of CV-OPS is to estimate, in a constant number of episodes, a strictly feasible policy and its associated violation, and then run S-OPS with such estimates. Algorithm 4 needs access to two anytime no-regret algorithms, one for adversarial MDPs with bandit feedback

that learns an estimated strictly feasible policy and one for the full-feedback setting on the simplex, which learns the most violated constraint. Specifically, CV-OPS employs an anytime regret minimizer for adversarial MDPs—called the primal algorithm $\mathcal{A}^P$—that attains, with probability at least $1 - C_P^\delta \delta$, for all $\tau \in [T], q \in \Delta(M)$, and for any sequence of loss functions the following regret bound $\sum_{t=1}^{\tau} \ell_t^\top (q_t - q) \leq C_{\mathcal{A}}^P \sqrt{\tau \ln(\tau)}$, where $C_{\mathcal{A}}^P$ encompass constant terms. This kind of guarantees are attained by state-of-the-arts algorithms for adversarial MDPs (e.g., (Jin et al., 2020)) after applying a standard doubling trick (Lattimore & Szepesvári, 2020). Algorithm 4 also employs an anytime online linear optimizer—called the dual algorithm $\mathcal{A}^D$—that attains for all $\tau \in [T], \phi \in \Delta_m$, and for any sequence of loss functions the following regret bound $\sum_{t=1}^{\tau} \ell_t^\top (\phi_t - \phi) \leq C_{\mathcal{A}}^D \sqrt{\tau}$, where $C_{\mathcal{A}}^D$ encompasses constant terms. This bound can be easily obtained by an online gradient descent algorithm (Orabona, 2019).

At each episode $t \in [T]$, Algorithm 4 requests a policy and a distribution over the $m$ constraints to $\mathcal{A}^P$ and $\mathcal{A}^D$, respectively (Lines 2-3). Thus, the algorithm plays the policy received by $\mathcal{A}^P$ and observes the usual bandit feedback for CMDPs (Line 4). Next, the loss functions for both the primal and the dual algorithm are built. Specifically, $\mathcal{A}^P$ receives the violation attained by the policy $\pi_t$ where any constraint is weighted given $\phi_t$ (Line 5), while $\mathcal{A}^D$ receives the negative of the violation attained for all $i \in [m]$ (Line 6). The estimation phase stops when the violation attained by the algorithm exceeds twice the regret bounds attained by both the primal and the dual algorithm plus the uncertainty on the estimation (Line 7). This condition is suitably chosen to ensure that the number of episodes are sufficient to estimate an approximation of $\pi^\diamond$, while still being constant. After the estimation, Algorithm 4 computes pessimistically the estimated Slater's parameter $\widehat{\rho}$ as the average violation attained during the estimation phase minus a quantity associated with the uncertainty of the estimation (Line 11). Finally, CV-OPS computes the strictly feasible solution as the uniform policy with respect to all the policies played in the estimation phase (Line 12) and runs S-OPS with $\beta_i = \alpha_i - \widehat{\rho}$, for all $i \in [m]$ and $\pi^\diamond = \widehat{\pi}^\diamond$ in input (Line 13).

### 6.1. Cumulative positive constraints violation

First, we show that the stopping condition at Line 7 allows to run the estimation phase for no more than a constant number of episodes. This is done in the following lemma.

**Lemma 6.1.** *Given any $\delta \in (0, 1)$, the episodes that Algorithm 4 uses to compute $\widehat{\rho}$ and $\widehat{\pi}^\diamond$ are $\bar{t} \leq 1/\rho^4 (3 C_{\mathcal{A}}^P + 10 L \ln \frac{1}{\delta} + 3 C_{\mathcal{A}}^D + L)^4$, with prob. at least $1 - (C_P^\delta + 2)\delta$.*

The bound could be reduced to $\bar{t} \leq 1/\rho^2 (3 C_{\mathcal{A}}^P + 10 L \ln \frac{1}{\delta} + 3 C_{\mathcal{A}}^D + L)^2$, with access to a no-regret algorithm without the logarithmic dependence on $T$ in the bound. Next, we

show that Algorithm 4 estimates a strictly feasible policy whose constraints margin is in $[\widehat{\rho}, \rho]$, as follows.

**Lemma 6.2.** *Given any $\delta \in (0, 1)$, Algorithm 4 guarantees $\widehat{\rho} \leq \min_{i \in [m]} (\alpha_i - \bar{g}_i^\top q^{P, \widehat{\pi}^\diamond}) \leq \rho$ with prob. at least $1 - 2\delta$.*

Finally, we provide the result in term of cumulative positive constraints violation attained by our algorithm.

**Theorem 6.3.** *Given $\delta \in (0, 1)$, Algorithm 4 attains $V_T \leq \mathcal{O}(L/\rho^4 (C_{\mathcal{A}}^P + L\sqrt{\ln \frac{1}{\delta}} + C_{\mathcal{A}}^D + L)^4)$ with probability at least $1 - (C_P^\delta + 7)\delta$.*

Theorem 6.3 is proved by employing Lemma 6.1 to bound the episodes in the estimation phase and Lemma 6.2 to state that S-OPS with $\widehat{\rho}, \widehat{\pi}^\diamond$ is safe with high probability.

### 6.2. Cumulative regret

We provide the theoretical guarantees attained by Algorithm 4 in terms of cumulative regret. To do so, we show that $\widehat{\rho}$ is not too small. This is done in the following lemma.

**Lemma 6.4.** *Given any $\delta \in (0, 1)$, Algorithm 4 guarantees $\widehat{\rho} \geq \rho/2$ with probability at least $1 - (C_P^\delta + 2)\delta$.*

Finally, we state the regret attained by CV-OPS.

**Theorem 6.5.** *Given any $\delta \in (0, 1)$, with $\eta = \gamma = \sqrt{L \ln(L|X||A|/\delta)/T|X||A|}$, Algorithm 4 attains regret $R_T \leq \mathcal{O}(\Theta L^3 |X| \sqrt{|A|T \ln(T|X||A|m/\delta)} + L/\rho^4 (C_{\mathcal{A}}^P + L \ln 1/\delta + C_{\mathcal{A}}^D + L)^4)$ with probability at least $1 - (C_P^\delta + 13)\delta$, where we let $\Theta := 1/\min\{\rho, \rho^2\}$.*

As Theorem 6.3, Theorem 6.5 is proved by employing Lemma 6.1 to bound the episodes in the estimation phase. Then, the result follows from the regret guarantees of S-OPS and Lemma 6.4 for $1/\widehat{\rho} \leq 2/\rho$. Differently from S-OPS, the bound of CV-OPS scales as the inverse of the Slater's parameter $\rho$, not as the (possibly smaller) margin of a generic strictly feasible policy. Thus, the bound of Algorithm 4 is asymptotically smaller than the one of S-OPS.

### 6.3. Lower bound on the regret

We conclude by showing that a dependency on the feasibility of the strictly feasible solution in the regret bound is unavoidable to guarantee violation of order $o(\sqrt{T})$, *which is the case of both the second and third setting*.

This is done by means of the following lower bound.

**Theorem 6.6.** *There exist two instances of CMDPs (with a single state and one constraint) such that, if in the first instance an algorithm suffers from a violation $V_T = o(\sqrt{T})$ probability at least $1 - n\delta$ for any $\delta \in (0, 1)$ and $n > 0$, then, in the second instance, it must suffer from a regret $R_T = \Omega(\frac{1}{\rho} \sqrt{T})$ with probability $3/4 - n\delta$.*

Notice that this lower bound holds for *any* algorithm attaining a violation bound that is $o(\sqrt{T})$, thus, it is still applicable to settings where the violations are allowed to be much larger than the ones attained by Algorithm 3 and Algorithm 4.

## Acknowledgments

This paper is supported by the Italian MIUR PRIN 2022 Project "Targeted Learning Dynamics: Computing Efficient and Fair Equilibria through No-Regret Algorithms", by the FAIR (Future Artificial Intelligence Research) project, funded by the NextGenerationEU program within the PNRR-PE-AI scheme (M4C2, Investment 1.3, Line on Artificial Intelligence), and by the EU Horizon project ELIAS (European Lighthouse of AI for Sustainability, No. 101120237).

## Impact Statement

This paper presents theoretical results and has the goal of advancing the field of Machine Learning. There are no potential societal consequences of our work which we feel must be highlighted here.

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

# Appendix

The appendix is organized as follows:

## A. Related Works

Online learning (Cesa-Bianchi & Lugosi, 2006; Orabona, 2019) in MDPs has received growing attention over the last years (see, *e.g.*, (Auer et al., 2008; Even-Dar et al., 2009; Neu et al., 2010)). Two types of feedback are usually investigated: *full feedback*, with the entire loss function being observed by the leaner, and *bandit feedback*, where the learner only observes losses of chosen actions. Azar et al. (2017) study learning in episodic MDPs with unknown transitions and stochastic losses under bandit feedback, achieving $\widetilde{\mathcal{O}}(\sqrt{T})$ regret, matching the lower bound for these MDPs. Rosenberg & Mansour (2019b) study learning under full feedback in episodic MDPs with adversarial losses and unknown transitions, achieving $\widetilde{\mathcal{O}}(\sqrt{T})$ regret. The same setting is studied by Rosenberg & Mansour (2019a) under bandit feedback, obtaining a suboptimal $\widetilde{\mathcal{O}}(T^{3/4})$ regret. Jin et al. (2020) provide an algorithm with an optimal $\widetilde{\mathcal{O}}(\sqrt{T})$ regret, in the same setting.

Online learning in CMDPs has generally been studied with stochastic losses and constraints. Zheng & Ratliff (2020) deal with fully-stochastic episodic CMDPs, assuming known transitions and bandit feedback. The regret of their algorithm is $\widetilde{\mathcal{O}}(T^{3/4})$, while its cumulative constraints violation is guaranteed to be below a threshold with a given probability. Bai et al. (2023) provide the first algorithm that achieves sublinear regret with unknown transitions, assuming that the rewards are deterministic and the constraints are stochastic with a particular structure. Efroni et al. (2020) propose two approaches to deal with the exploration-exploitation trade-off in episodic CMDPs. The first one resorts to a linear programming formulation of CMDPs and obtains sublinear regret and cumulative positive constraints violation. The second one relies on a primal-dual formulation of the problem and guarantees sublinear regret and cumulative (positive/negative) constraints violation, when transitions, losses, and constraints are unknown and stochastic, under bandit feedback. Liu et al. (2021); Müller et al. (2024); Stradi et al. (2025) study stochastic *hard* constraints; however, the authors only focus on stochastic losses. Bura et al. (2022) focus on our second scenario and develop a pessimist algorithm for the stochastic setting only. In this work, the strictly safe policy is played for a certain amount of time, after which a good estimate on constraints costs and transitions is available. This allows the pessimistic set to be large enough to to be used. We underline that, optimizing over the pessimistic decision space cannot be done in our setting, since adversarial no-regret algorithms do not work in increasing decision spaces. Recently, Shi et al. (2023) study stochastic hard constraints on both states and actions. As concerns adversarial settings, Wei et al. (2018); Qiu et al. (2020); Stradi et al. (2024a) address CMDPs with adversarial losses, but they only provide guarantees in terms of *soft* constraints. Moreover, Wei et al. (2023); Ding & Lavaei (2023); Stradi et al. (2024b) consider non-stationary losses/constraints with bounded variation. Their results do *not* apply to general adversarial losses.

Hard constraints have also been studied in online convex optimization (Guo et al., 2022), and in stochastic settings with simpler structure (Chen et al., 2018; Pacchiano et al., 2021; Bernasconi et al., 2022). Our results are much more general than those, as we jointly consider adversarial losses, bandit feedback, and an MDP structure.

## B. Omitted proofs for the clean event

In this section, we report the omitted proof related to the clean event. We start stating the following preliminary result.

**Lemma B.1.** *Given any $\delta \in (0, 1)$, fix $i \in [m]$, $t \in [T]$ and $(x, a) \in X \times A$, it holds, with probability at least $1 - \delta$:*

$$\left| \widehat{g}_{t,i}(x, a) - \overline{g}_i(x, a) \right| \leq \zeta_t(x, a),$$

*where $\zeta_t(x, a) := \sqrt{\frac{\ln(2/\delta)}{2N_t(x,a)}}$ and $\overline{g}_{t,i}(x, a)$ is the true mean value of the distribution.*

*Proof.* Focus on specifics $i \in [m]$, $t \in [T]$ and $(x, a) \in X \times A$. By Hoeffding's inequality and noticing that constraints values are bounded in $[0, 1]$, it holds that:

$$\mathbb{P}\left[ \left| \widehat{g}_{t,i}(x, a) - \overline{g}_i(x, a) \right| \geq \frac{c}{N_t(x, a)} \right] \leq 2 \exp\left( -\frac{2c^2}{N_t(x, a)} \right)$$

Setting $\delta = 2 \exp\left( -\frac{2c^2}{N_t(x,a)} \right)$ and solving to find a proper value of $c$ concludes the proof. $\square$

Now we generalize the previous result in order to hold for every $i \in [m]$, $t \in [T]$ and $(x, a) \in X \times A$ at the same time.

**Lemma B.2.** *Given a confidence $\delta \in (0, 1)$, with probability at least $1 - \delta$, for every $i \in [m]$, episode $t \in [T]$, and pair $(x, a) \in X \times A$, it holds $|\widehat{g}_{t,i}(x, a) - \overline{g}_i(x, a)| \leq \xi_t(x, a)$, where we let the confidence bound $\xi_t(x, a) := \min\{1, \sqrt{4 \ln(T|X||A|m/\delta)/\max\{1, N_t(x, a)\}}\}$.*

*Proof.* From Lemma G.1, given $\delta' \in (0, 1)$, we have for any $i \in [m]$, $t \in [T]$ and $(x, a) \in X \times A$:

$$\mathbb{P}\left[ \left| \widehat{g}_{t,i}(x, a) - \overline{g}_i(x, a) \right| \leq \zeta_t(x, a) \right] \geq 1 - \delta'.$$

Now, we are interested in the intersection of all the events, namely,

$$\mathbb{P}\left[ \bigcap_{x,a,m,t} \left\{ \left| \widehat{g}_{t,i}(x, a) - \overline{g}_i(x, a) \right| \leq \zeta_t(x, a) \right\} \right].$$

Thus, we have:

$$\mathbb{P}\left[ \bigcap_{x,a,m,t} \left\{ \left| \widehat{g}_{t,i}(x, a) - \overline{g}_i(x, a) \right| \leq \zeta_t(x, a) \right\} \right]$$

$$= 1 - \mathbb{P}\left[ \bigcup_{x,a,m,t} \left\{ \left| \widehat{g}_{t,i}(x, a) - \overline{g}_i(x, a) \right| \leq \zeta_t(x, a) \right\}^c \right]$$

$$\geq 1 - \sum_{x,a,m,t} \mathbb{P}\left[ \left\{ \left| \widehat{g}_{t,i}(x, a) - \overline{g}_i(x, a) \right| \leq \zeta_t(x, a) \right\}^c \right] \tag{3}$$

$$\geq 1 - |X||A|mT\delta',$$

where Inequality (3) holds by Union Bound. Noticing that $g_{t,i}(x, a) \leq 1$, substituting $\delta'$ with $\delta := \delta'/|X||A|mT$ in $\zeta_t(x, a)$ with an additional Union Bound over the possible values of $N_t(x, a)$, and thus obtaining $\xi_t(x, a)$, concludes the proof. $\square$

## C. Omitted proofs for sublinear violation

In this section we report the omitted proofs of the theoretical results for Algorithm 2.

## C.1. Feasibility

We start by showing that Program (2) admits a feasible solution with arbitrarily large probability.

**Lemma 4.1.** *Given confidence $\delta \in (0,1)$, Algorithm 2 ensures that* $\mathrm{PROJ}(\tilde{q}_{t+1}, \widehat{G}_t, \Xi_t, \mathcal{P}_t)$ *is feasible at every episode* $t \in [T]$ *with probability at least* $1 - 5\delta$.

*Proof.* To prove the lemma we show that under the event $\mathcal{E}^{G,\Delta}(\delta)$, which holds the probability at least $1 - 5\delta$, Program (2) admits a feasible solution. Precisely, under the event $\mathcal{E}^{\Delta}(\delta)$, the true transition function $P$ belongs to $\mathcal{P}_t$ at each episode. Moreover, under the event $\mathcal{E}^G(\delta)$, we have, for any feasible solution $q^{\square}$ of the offline optimization problem, for any $t \in [T]$,

$$\left(\widehat{G}_t - \Xi_t\right)^{\top} q^{\square} \preceq \overline{G}_t^{\top} q^{\square} \preceq \alpha,$$

where the first inequality holds by the definition of the event. The previous inequality shows that if $q^{\square}$ satisfies the constraints with respect to the true mean constraint matrix, it satisfies also the optimistic constraints. Thus, the feasible solutions to the offline problem are all available at every episode. Noticing that the clean event is defined as the intersection between $\mathcal{E}^G(\delta)$ and $\mathcal{E}^{\Delta}(\delta)$ concludes the proof. $\square$

## C.2. Violations

We proceed bounding the cumulative positive violation as follows.

**Theorem 4.2.** *Given $\delta \in (0,1)$, Algorithm 2 attains* $V_T \leq \mathcal{O}\left(L|X|\sqrt{|A|T \ln\left(T|X||A|m/\delta\right)}\right)$ *with prob. at least* $1 - 8\delta$.

*Proof.* The key point of the problem is to relate the constraints satisfaction with the convergence rate of both the confidence bound on the constraints and the transitions.

First, we notice that under the clean event $\mathcal{E}^{G,\Delta}(\delta)$, all the following reasoning hold for every constraint $i \in [m]$. Thus, we focus on the bound of a single constraint violation problem defined as follows:

$$V_T := \sum_{t=1}^{T} \left[\overline{g}^{\top} q_t - \alpha\right]^+$$

By Lemma 4.1, under the clean event the $\mathcal{E}^{G,\Delta}(\delta)$, the convex program is feasible and it holds:

$$\overline{g} - 2\xi_t \preceq \widehat{g}_t - \xi_t$$

Thus, multiplying for the estimated occupancy measure and by construction of the convex program we obtain:

$$(\overline{g} - 2\xi_{t-1})^{\top} \widehat{q}_t \leq (\widehat{g}_{t-1} - \xi_{t-1})^{\top} \widehat{q}_t \leq \alpha.$$

Rearranging the equation, it holds:

$$\overline{g}^{\top} \widehat{q}_t \leq \alpha + 2\xi_{t-1}^{\top} \widehat{q}_t.$$

Now, in order to obtain the instantaneous violation definition we proceed as follows,

$$\overline{g}^{\top} \widehat{q}_t + \overline{g}^{\top} q_t - \overline{g}^{\top} q_t \leq \alpha + 2\xi_{t-1}^{\top} \widehat{q}_t,$$

from which we obtain:

$$\overline{g}^{\top} q_t - \alpha \leq \overline{g}^{\top} (q_t - \widehat{q}_t) + 2\xi_{t-1}^{\top} \widehat{q}_t$$
$$\leq \|\overline{g}\|_{\infty} \|q_t - \widehat{q}_t\|_1 + 2\xi_{t-1}^{\top} \widehat{q}_t,$$

where the last step holds by the Hölder inequality. Notice that, since the RHS of the previous inequality is greater than zero, it holds,

$$[\overline{g}^{\top} q_t - \alpha]^+ \leq \|q_t - \widehat{q}_t\|_1 + 2\xi_{t-1}^{\top} \widehat{q}_t.$$

which leads to $V_T \leq \sum_{t=1}^{T} \|q_t - \widehat{q}_t\|_1 + 2\sum_{t=1}^{T} \xi_{t-1}^{\top} \widehat{q}_t$, where the first part of the equation refers to the estimate of the transitions while the second one to the estimate of the constraints. We will bound the two terms separately.

**Bound on $\sum_{t=1}^{T}\|\widehat{q}_t - q_t\|_1$.** The term of interest encodes the distance between the estimated occupancy measure and the real one chosen by the algorithm. Thus, it depends on the estimation of the true transition functions. To bound the quantity of interest, we proceed as follows:

$$\sum_{t=1}^{T}\|\widehat{q}_t - q_t\|_1 = \sum_{t=1}^{T}\sum_{x,a}|\widehat{q}_t(x,a) - q_t(x,a)|$$

$$\leq \mathcal{O}\left(L|X|\sqrt{|A|T\ln\left(\frac{T|X||A|}{\delta}\right)}\right),\tag{4}$$

where Inequality (4) holds since, by Lemma G.2, under the clean event, with probability at least $1-2\delta$, we have $\sum_{t=1}^{T}\sum_{x,a}|\widehat{q}_t(x,a) - q_t(x,a))| \leq \mathcal{O}\left(L|X|\sqrt{|A|T\ln\left(\frac{T|X||A|}{\delta}\right)}\right)$, when $\widehat{q}_t \in \Delta(\mathcal{P}_t)$. Please notice that the condition $\widehat{q}_t \in \Delta(\mathcal{P}_t)$ is verified since the constrained space defined by Program (2) is contained in $\Delta(\mathcal{P}_t)$.

**Bound on $\sum_{t=1}^{T}\xi_{t-1}^{\top}\widehat{q}_t$.** This term encodes the estimation of the constraints functions obtained following the estimated occupancy measure. Nevertheless, since the confidence bounds converge only for the paths traversed by the learner, it is necessary to relate $\xi_t$ to the real occupancy measure chosen by the algorithm. To do so, we notice that by Hölder inequality and since $\xi_t(x,a) \leq 1$, it holds:

$$\sum_{t=1}^{T}\xi_{t-1}^{\top}\widehat{q}_t \leq \sum_{t=1}^{T}\xi_{t-1}^{\top}q_t + \sum_{t=1}^{T}\xi_{t-1}^{\top}(\widehat{q}_t - q_t)$$

$$\leq \sum_{t=1}^{T}\xi_{t-1}^{\top}q_t + \sum_{t=1}^{T}\|\xi_{t-1}\|_{\infty}\|\widehat{q}_t - q_t\|_1$$

$$\leq \sum_{t=1}^{T}\xi_{t-1}^{\top}q_t + \sum_{t=1}^{T}\|\widehat{q}_t - q_t\|_1.$$

The second term of the inequality is bounded by the previous analysis, while for the first term we proceed as follows:

$$\sum_{t=1}^{T}\xi_{t-1}^{\top}q_t = \sum_{t=1}^{T}\sum_{x,a}\xi_{t-1}(x,a)q_t(x,a)$$

$$\leq \sum_{t=1}^{T}\sum_{x,a}\xi_{t-1}(x,a)\mathbb{1}_t\{x,a\} + L\sqrt{2T\ln\frac{1}{\delta}}\tag{5}$$

$$= \sqrt{4\ln\left(\frac{T|X||A|m}{\delta}\right)}\sum_{t=1}^{T}\sum_{x,a}\sqrt{\frac{1}{\max\{1,N_{t-1}(x,a)\}}}\mathbb{1}_t\{x,a\} + L\sqrt{2T\ln\frac{1}{\delta}}$$

$$\leq 3\sqrt{4\ln\left(\frac{T|X||A|m}{\delta}\right)}\sum_{x,a}\sqrt{N_T(x,a)} + L\sqrt{2T\ln\frac{1}{\delta}}\tag{6}$$

$$\leq 6\sqrt{L|X||A|T\ln\left(\frac{T|X||A|m}{\delta}\right)} + L\sqrt{2T\ln\frac{1}{\delta}},\tag{7}$$

where Inequality (5) follows from Azuma inequality and noticing that $\sum_{x,a}\xi_{t-1}(x,a)q_t(x,a) \leq L$ (with probability at least $1-\delta$), Inequality (6) holds since $1 + \sum_{t=1}^{T}\frac{1}{\sqrt{t}} \leq 2\sqrt{T} + 1 \leq 3\sqrt{T}$ and Inequality (7) follows from Cauchy-Schwarz inequality and noticing that $\sqrt{\sum_{x,a}N_T(x,a)} \leq \sqrt{LT}$.

We combine the previous bounds as follows:

$$V_T \leq \sum_{t=1}^{T}\|q_t - \widehat{q}_t\|_1 + 2\sum_{t=1}^{T}\xi_{t-1}^{\top}\widehat{q}_t$$

$$\leq \mathcal{O}\left(L|X|\sqrt{|A|T\ln\left(\frac{T|X||A|m}{\delta}\right)}\right).$$

The results holds with probability at least at least $1 - 8\delta$ by union bound over the clean event, Lemma G.2 and the Azuma-Hoeffding inequality. This concludes the proof. □

### C.3. Regret

In this section, we prove the regret bound of Algorithm 2. Precisely, the bound follows from noticing that, under the clean event, the optimal safe solution is included in the decision space for every episode $t \in [T]$.

**Theorem 4.3.** *Given* $\delta \in (0,1)$, *by setting* $\eta = \gamma = \sqrt{L\ln(L|X||A|/\delta)/T|X||A|}$, *Algorithm* 2 *attains* $R_T \leq \mathcal{O}\left(L|X|\sqrt{|A|T\ln\left(T|X||A|/\delta\right)}\right)$ *with prob. at least* $1 - 10\delta$.

*Proof.* We first rewrite the regret definition as follows:

$$R_T = \sum_{t=1}^{T}\ell_t^\top q_t - \sum_{t=1}^{T}\ell_t^\top q^*$$

$$= \underbrace{\sum_{t=1}^{T}\ell_t^\top(q_t - \widehat{q}_t)}_{①} + \underbrace{\sum_{t=1}^{T}\widehat{\ell}_t^\top(\widehat{q}_t - q^*)}_{②} + \underbrace{\sum_{t=1}^{T}(\ell_t - \widehat{\ell}_t)^\top\widehat{q}_t}_{③} + \underbrace{\sum_{t=1}^{T}(\widehat{\ell}_t - \ell_t)^\top q^*}_{④}.$$

Precisely, the first term encompasses the distance between the true transitions and the estimated ones, the second concerns the optimization performed by online mirror descent and the last ones encompass the bias of the estimators.

**Bound on ①.** We start bounding the first term, namely, the cumulative distance between the estimated occupancy measure and the real one, as follows:

$$① = \sum_{t=1}^{T}\ell_t^\top(q_t - \widehat{q}_t)$$

$$= \sum_{t=1}^{T}\sum_{x,a}\ell_t(x,a)(q_t(x,a) - \widehat{q}_t(x,a))$$

$$\leq \sum_{t=1}^{T}\sum_{x,a}|(q_t(x,a) - \widehat{q}_t(x,a))|, \tag{8}$$

where the Inequality (8) holds by Hölder inequality noticing that $\|\ell_t\|_\infty \leq 1$ for all $t \in [T]$. Then, noticing that the projection of Algorithm 2 is performed over a subset of $\Delta(\mathcal{P}_t)$ and employing Lemma G.2, we obtain:

$$① \leq \mathcal{O}\left(L|X|\sqrt{|A|T\ln\left(\frac{T|X||A|}{\delta}\right)}\right), \tag{9}$$

with probability at least $1 - 2\delta$, under the clean event.

**Bound on ②.** To bound the second term, we underline that, under the clean event $\mathcal{E}^{G,\Delta}(\delta)$, the estimated safe occupancy $\widehat{q}_t$ belongs to $\Delta(\mathcal{P}_t)$ and the optimal safe solution $q^*$ is included in the constrained decision space for each $t \in [T]$. Moreover we notice that, for each $t \in [T]$, the constrained space is convex and linear, by construction of Program (2). Thus, following the standard analysis of online mirror descent (Orabona, 2019) and from Lemma G.6, we have, under the clean event:

$$② \leq \frac{L\ln\left(|X|^2|A|\right)}{\eta} + \eta\sum_{t,x,a}\widehat{q}_t(x,a)\widehat{\ell}_t(x,a)^2.$$

Thus, to bound the biased estimator, we notice that $\widehat{q}_t(x,a)\widehat{\ell}_t(x,a)^2 \leq \frac{\widehat{q}_t(x,a)}{u_t(x,a)+\gamma}\widehat{\ell}_t(x,a) \leq \widehat{\ell}_t(x,a)$. We then apply Lemma G.3 with $\alpha_t(x,a) = 2\gamma$ and obtain $\sum_{t,x,a} \widehat{q}_t(x,a)\widehat{\ell}_t(x,a)^2 \leq \sum_{t,x,a} \frac{q_t(x,a)}{u_t(x,a)}\ell_t(x,a) + \frac{L\ln\frac{L}{\delta}}{2\gamma}$. Finally, we notice that, under the clean event, $q_t(x,a) \leq u_t(x,a)$, obtaining, with probability at least $1 - \delta$:

$$② \leq \frac{L\ln\left(|X|^2|A|\right)}{\eta} + \eta|X||A|T + \frac{\eta L\ln(L/\delta)}{2\gamma}.$$

Setting $\eta = \gamma = \sqrt{\frac{L\ln(L|X||A|/\delta)}{T|X||A|}}$, we obtain:

$$② \leq \mathcal{O}\left(L\sqrt{|X||A|T\ln\left(\frac{|X||A|}{\delta}\right)}\right), \tag{10}$$

with probability at least $1 - \delta$, under the clean event.

**Bound on ③.** The third term follows from Lemma G.5, from which, under the clean event, with probability at least $1 - 3\delta$ and setting $\gamma = \sqrt{\frac{L\ln(L|X||A|/\delta)}{T|X||A|}}$, we obtain:

$$③ \leq \mathcal{O}\left(L|X|\sqrt{|A|T\ln\left(\frac{T|X||A|}{\delta}\right)}\right). \tag{11}$$

**Bound on ④.** We bound the fourth term employing Corollary G.4 and obtaining,

$$\sum_{t=1}^{T}\left(\widehat{\ell}_t - \ell_t\right)^\top q^* = \sum_{t,x,a} q^*(x,a)\left(\widehat{\ell}_t(x,a) - \ell_t(x,a)\right)$$

$$\leq \sum_{t,x,a} q^*(x,a)\ell_t(x,a)\left(\frac{q_t(x,a)}{u_t(x,a)} - 1\right) + \sum_{x,a} \frac{q^*(x,a)\ln\frac{|X||A|}{\delta}}{2\gamma}$$

$$= \sum_{t,x,a} q^*(x,a)\ell_t(x,a)\left(\frac{q_t(x,a)}{u_t(x,a)} - 1\right) + \frac{L\ln\frac{|X||A|}{\delta}}{2\gamma}.$$

Noticing that, under the clean event, $q_t(x,a) \leq u_t(x,a)$ and setting $\gamma = \sqrt{\frac{L\ln(L|X||A|/\delta)}{T|X||A|}}$, we obtain, with probability at least $1 - \delta$:

$$④ \leq \mathcal{O}\left(L\sqrt{|X||A|T\ln\left(\frac{T|X||A|}{\delta}\right)}\right). \tag{12}$$

**Final result.** Finally, combining Equation (9), Equation (10), Equation (11) and Equation (12) and applying a union bound, we obtain, with probability at least $1 - 10\delta$,

$$R_T \leq \mathcal{O}\left(L|X|\sqrt{|A|T\ln\left(\frac{T|X||A|}{\delta}\right)}\right).$$

$\square$

# D. Omitted proofs when Condition 2.5 holds

In this section we report the omitted proofs of the theoretical results for Algorithm 3.

## D.1. Safety

We start by showing that Algorithm 3 is safe with high probability.

**Theorem 5.1.** *Given a confidence $\delta \in (0,1)$, Algorithm 3 is safe with probability at least $1 - 5\delta$.*

*Proof.* We show that, under event $\mathcal{E}^{G,\Delta}(\delta)$, the *non-Markovian* policy defined by the probability $\lambda_t$ satisfies the constraints. Intuitively, the result follows from the construction of the convex combination parameter $\lambda_t$. Indeed, $\lambda_t$ is built using a pessimist estimated of the constraints cost, namely, $\widehat{g}_{t,i} + \xi_t$. Moreover, the upper occupancy bound $\widehat{u}_t$ introduces pessimism in the choice of the transition function. Finally, the $\max_{i \in [m]}$ operator allows to be conservative for all the $m$ constraints.

We split the analysis in the two possible cases defined by $\lambda_t$, namely, $\lambda_t = 0$ and $\lambda_t \in (0,1)$. Please notice that $\lambda_t < 1$, by construction.

**Analysis when $\lambda_t = 0$.** When $\lambda_t = 0$, it holds, by construction, that $\forall i \in [m] : (\widehat{g}_{t-1,i} + \xi_{t-1})^\top \widehat{u}_t \leq \alpha_i$. Thus, under the event $\mathcal{E}^{G,\Delta}(\delta)$, it holds, $\forall i \in [m]$:

$$\alpha_i \geq (\widehat{g}_{t-1,i} + \xi_{t-1})^\top \widehat{u}_t$$
$$\geq (\widehat{g}_{t-1,i} + \xi_{t-1})^\top \widehat{q}_t \tag{13}$$
$$= (\widehat{g}_{t-1,i} + \xi_{t-1})^\top q_t$$
$$\geq \overline{g}_i^\top q_t, \tag{14}$$

where Inequality (13) holds by definition of $\widehat{u}_t$ and Inequality (14) by the pessimistic definition of the constraints.

**Analysis when $\lambda_t \in (0,1)$.** We focus on a single constraint $i \in [m]$, then we generalize the analysis for the entire set of constraints. First we notice that the constraints cost, for a single constraint $i \in [m]$, attained by the *non-Markovian* policy $\pi_t$, is equal to $\lambda_{t-1} \overline{g}_i^\top q^\diamond + (1 - \lambda_{t-1}) \overline{g}_i^\top q^{P,\widehat{\pi}_t}$. Thus, it holds by definition of the known strictly feasible $\pi^\diamond$,

$$\lambda_{t-1} \overline{g}_i^\top q^\diamond + (1 - \lambda_{t-1}) \overline{g}_i^\top q^{P,\widehat{\pi}_t} = \lambda_{t-1} \beta_i + (1 - \lambda_{t-1}) \overline{g}_i^\top q^{P,\widehat{\pi}_t}. \tag{15}$$

Then, we consider both the cases when $L < (\widehat{g}_{t-1,i} + \xi_{t-1})^\top \widehat{u}_t$ (first case) and $L > (\widehat{g}_{t-1,i} + \xi_{t-1})^\top \widehat{u}_t$ (second case). If the two quantities are equivalent, the proof still holds breaking the ties arbitrarily.

*First case.* It holds that:

$$\lambda_{t-1} \beta_i + (1 - \lambda_{t-1}) \overline{g}_i^\top q^{\widehat{\pi}_t, P} \leq \lambda_{t-1} \beta_i + (1 - \lambda_{t-1}) L \tag{16}$$
$$= \frac{L - \alpha_i}{L - \beta_i}(\beta_i - L) + L$$
$$= \frac{\alpha_i - L}{\beta_i - L}(\beta_i - L) + L$$
$$= \alpha_i,$$

where Inequality (16) holds by definition of the constraints.

*Second case.* It holds that:

$$\lambda_{t-1} \beta_i + (1 - \lambda_{t-1}) \overline{g}_i^\top q^{P,\widehat{\pi}_t}$$
$$\leq \lambda_{t-1} \beta_i + (1 - \lambda_{t-1})(\widehat{g}_{t-1,i} + \xi_{t-1})^\top q^{P,\widehat{\pi}_t} \tag{17}$$
$$\leq \lambda_{t-1} \beta_i + (1 - \lambda_{t-1})(\widehat{g}_{t-1,i} + \xi_{t-1})^\top \widehat{u}_t \tag{18}$$
$$= \lambda_{t-1} \beta_i - \lambda_{t-1}(\widehat{g}_{t-1,i} + \xi_{t-1})^\top \widehat{u}_t + (\widehat{g}_{t-1,i} + \xi_{t-1})^\top \widehat{u}_t$$
$$= \lambda_{t-1}(\beta_i - (\widehat{g}_{t-1,i} + \xi_{t-1})^\top \widehat{u}_t) + (\widehat{g}_{t-1,i} + \xi_{t-1})^\top \widehat{u}_t$$
$$\leq \frac{(\widehat{g}_{t-1,i} + \xi_{t-1})^\top \widehat{u}_t - \alpha_i}{(\widehat{g}_{t-1,i} + \xi_{t-1})^\top \widehat{u}_t - \beta_i}(\beta_i - (\widehat{g}_{t-1,i} + \xi_{t-1})^\top \widehat{u}_t) + (\widehat{g}_{t-1,i} + \xi_{t-1})^\top \widehat{u}_t$$
$$= \frac{\alpha_i - (\widehat{g}_{t-1,i} + \xi_{t-1})^\top \widehat{u}_t}{\beta_i - (\widehat{g}_{t-1,i} + \xi_{t-1})^\top \widehat{u}_t}(\beta_i - (\widehat{g}_{t-1,i} + \xi_{t-1})^\top \widehat{u}_t) + (\widehat{g}_{t-1,i} + \xi_{t-1})^\top \widehat{u}_t$$
$$= \alpha_i - (\widehat{g}_{t-1,i} + \xi_{t-1})^\top \widehat{u}_t + (\widehat{g}_{t-1,i} + \xi_{t-1})^\top \widehat{u}_t$$

$$= \alpha_i,$$

where Inequality (17) holds by the definition of the event and Inequality (18) holds by the definition of $\widehat{u}_t$.

To conclude the proof, we underline that $\lambda_t$ is chosen taking the maximum over the constraints, which implies that the more conservative $\lambda_t$ (the one which takes the combination nearer to the strictly feasible solution) is chosen. Thus, all the constraints are satisfied. □

### D.2. Regret

We start by the statement of the following Lemma, which is a generalization of the results from (Jin et al., 2020). Intuitively, the following result states that the distance between the estimated *non-safe* occupancy measure $\widehat{q}_t$ and the real one reduces as the number of episodes increases, paying a $1 - \lambda_t$ factor. This is reasonable since, from the update of the *non-Markovian* policy $\pi_t$ (see Algorithm 3), policy $\widehat{\pi}_t \leftarrow \widehat{q}_t$ is played with probability $1 - \lambda_{t-1}$.

**Lemma D.1.** *Under the clean event, with probability at least $1 - 2\delta$, for any collection of transition functions $\{P_t^x\}_{x \in X}$ such that $P_t^x \in \mathcal{P}_t$, and for any collection of $\{\lambda_t\}_{t=0}^{T-1}$ used to select policy $\pi_{t+1}$, we have, for all $x$,*

$$\sum_{t=1}^{T}(1 - \lambda_{t-1}) \sum_{x \in X, a \in A} \left| q^{P_t^x, \widehat{\pi}_t}(x,a) - q^{P, \widehat{\pi}_t}(x,a) \right| \leq \mathcal{O}\left( L|X| \sqrt{|A|T \ln\left(\frac{T|X||A|}{\delta}\right)} \right).$$

*Proof.* We will refer as $q_t^x$ to $q^{P_t^x, \pi_t}$ and as $\widehat{q}_t^x$ to $q^{P_t^x, \widehat{\pi}_t}$. Moreover, we define:

$$\epsilon_t^*(x'|x,a) = \sqrt{\frac{P(x'|x,a)\ln\left(\frac{T|X||A|}{\delta}\right)}{\max\{1, N_t(x,a)\}}} + \frac{\ln\left(\frac{T|X||A|}{\delta}\right)}{\max\{1, N_t(x,a)\}}.$$

Now following standard analysis by Lemma G.2 from (Jin et al., 2020), we have that,

$$\sum_{t=1}^{T}(1 - \lambda_{t-1}) \sum_{x \in X, a \in A} \left| q^{P_t^x, \widehat{\pi}_t}(x,a) - q^{P, \widehat{\pi}_t}(x,a) \right| \leq$$

$$\sum_{0 \leq m < k < L} \sum_{t, w_m} (1 - \lambda_{t-1})\epsilon_t^*(x_{m+1}|x_m, a_m)q^{P, \widehat{\pi}_t}(x_m, a_m) + |X| \sum_{0 \leq m < h < L} \sum_{t, w_m, w_h'} (1 - \lambda_{t-1}) \cdot$$

$$\cdot \, \epsilon_{i_t}^*(x_{m+1} \mid x_m, a_m)\, q^{P, \widehat{\pi}_t}(x_m, a_m)\, \epsilon_t^*(x_{h+1}' \mid x_h', a_h')\, q^{P, \widehat{\pi}_t}(x_h', a_h' \mid x_{m+1}),$$

where $w_m = (x_m, a_m, x_{m+1})$.

**Bound on the first term.** To bound the first term we notice that, by definition of $q^{P, \widehat{\pi}_t}$ it holds:

$$\sum_{0 \leq m < k < L} \sum_{t, w_m} (1 - \lambda_{t-1})\epsilon_t^*(x_{m+1}|x_m, a_m)q^{P, \widehat{\pi}_t}(x_m, a_m)$$

$$= \sum_{0 \leq m < k < L} \sum_{t, w_m} \epsilon_t^*(x_{m+1}|x_m, a_m) \left( q^{P, \pi_t}(x_m, a_m) - \lambda_{t-1}q^{P, \pi^\diamond}(x_m, a_m) \right)$$

$$\leq \sum_{0 \leq m < k < L} \sum_{t, w_m} \epsilon_t^*(x_{m+1}|x_m, a_m)q^{P, \pi_t}(x_m, a_m)$$

$$\leq \mathcal{O}\left( L|X| \sqrt{|A|T \ln\left(\frac{T|X||A|}{\delta}\right)} \right),$$

where the last step holds following Lemma G.2 from (Jin et al., 2020).

**Bound on the second term.** Following Lemma G.2 from (Jin et al., 2020), the second term is bounded by (ignoring constants),

$$\sum_{0 \le m < h < L} \sum_{t, w_m, w'_h} (1 - \lambda_{t-1}) \sqrt{\frac{P(x_{m+1} \mid x_m, a_m) \ln\left(\frac{T|X||A|}{\delta}\right)}{\max\{1, N_t(x_m, a_m)\}}}.$$

$$\cdot q^{P, \widehat{\pi}_t}(x_m, a_m) \sqrt{\frac{P(x'_{h+1} \mid x'_h, a'_h) \ln\left(\frac{T|X||A|}{\delta}\right)}{\max\{1, N_t(x'_h, a'_h)\}}} q^{P, \widehat{\pi}_t}(x'_h, a'_h \mid x_{m+1})$$

$$+ \sum_{0 \le m < h < L} \sum_{t, w_m, w'_h} (1 - \lambda_{t-1}) \frac{q^{P, \widehat{\pi}_t}(x_m, a_m) \ln\left(\frac{T|X||A|}{\delta}\right)}{\max\{1, N_t(x_m, a_m)\}} +$$

$$+ \sum_{0 \le m < h < L} \sum_{t, w_m, w'_h} (1 - \lambda_{t-1}) \frac{q^{P, \widehat{\pi}_t}(x'_h, a'_h) \ln\left(\frac{T|X||A|}{\delta}\right)}{\max\{1, N_t(x'_h, a'_h)\}}.$$

The last two terms are bounded logarithmically in $T$, employing the definition of $q^{P, \widehat{\pi}_t}$ and following Lemma G.2 from (Jin et al., 2020), while, similarly, the first term is bounded by:

$$\sum_{0 \le m < h < L} \sqrt{|X_{m+1}| \sum_{t, x_m, a_m} \frac{(1 - \lambda_{t-1}) q^{P, \widehat{\pi}_t}(x_m, a_m)}{\max\{1, N_t(x_m, a_m)\}}} \sqrt{|X_{h+1}| \sum_{t, x'_h, a'_h} \frac{(1 - \lambda_{t-1}) q^{P, \widehat{\pi}_t}(x'_h, a'_h)}{\max\{1, N_t(x'_h, a'_h)\}}},$$

which is upper bounded by:

$$\sum_{0 \le m < h < L} \sqrt{|X_{m+1}| \sum_{t, x_m, a_m} \frac{q_t(x_m, a_m)}{\max\{1, N_t(x_m, a_m)\}}} \sqrt{|X_{h+1}| \sum_{t, x'_h, a'_h} \frac{q_t(x'_h, a'_h)}{\max\{1, N_t(x'_h, a'_h)\}}}.$$

Employing the same argument as Lemma G.2 from (Jin et al., 2020) shows that the previous term is bounded logarithmically in $T$ and concludes the proof. $\square$

We are now ready to prove the regret bound attained by Algorithm 3.

**Theorem 5.2.** *Given* $\delta \in (0, 1)$, *by setting* $\eta = \gamma = \sqrt{L \ln(L|X||A|/\delta)/T|X||A|}$, *Algorithm 3 attains* $R_T \le \mathcal{O}\left(\Psi L^3 |X| \sqrt{|A|T \ln(T|X||A|m/\delta)}\right)$ *with prob. at least* $1 - 11\delta$, *where* $\Psi := \max_{i \in [m]}\{1/\min\{(\alpha_i - \beta_i), (\alpha_i - \beta_i)^2\}\}$.

*Proof.* We start decomposing the $R_T := \sum_{t=1}^{T} \ell_t^\top(q_t - q^*)$ definition as:

$$\underbrace{\sum_{t=1}^{T} \ell_t^\top\left(q_t - q^{P_t, \pi_t}\right)}_{①} + \underbrace{\sum_{t=1}^{T} \widehat{\ell}_t^\top\left(q^{P_t, \widehat{\pi}_t} - q^*\right)}_{②} + \underbrace{\sum_{t=1}^{T} \ell_t^\top\left(q^{P_t, \pi_t} - q^{P_t, \widehat{\pi}_t}\right)}_{③} +$$

$$+ \underbrace{\sum_{t=1}^{T} \left(\ell_t - \widehat{\ell}_t\right)^\top q^{P_t, \widehat{\pi}_t}}_{④} + \underbrace{\sum_{t=1}^{T} \left(\widehat{\ell}_t - \ell_t\right)^\top q^*}_{⑤},$$

where $P_t$ is the transition chosen by the algorithm at episode $t$. Precisely, the first term encompasses the estimation of the transition functions, the second term concerns the optimization performed by the algorithm, the third term encompasses the regret accumulated by performing the convex combination of policies and the last two terms concern the bias of the optimistic estimators.

We proceed bounding the five terms separately.

**Bound on** ① We bound the first term as follows:

$$\text{①} = \sum_{t=1}^{T} \ell_t^\top \left( q_t - q^{P_t, \pi_t} \right)$$

$$= \sum_{t=1}^{T} \sum_{x,a} \ell_t(x,a) \left( q_t(x,a) - q^{P_t, \pi_t}(x,a) \right)$$

$$\leq \sum_{t=1}^{T} \sum_{x,a} \left| q_t(x,a) - q^{P_t, \pi_t}(x,a) \right|,$$

where the last inequality holds by Hölder inequality noticing that $\|\ell_t\|_\infty \leq 1$ for all $t \in [T]$. Then we can employ Lemmas G.2, since $\pi_t$ is the policy that guides the exploration and $P_t \in \mathcal{P}_t$, obtaining:

$$\text{①} \leq \mathcal{O}\left( L|X|\sqrt{|A|T \ln\left(\frac{T|X||A|}{\delta}\right)} \right), \tag{19}$$

with probability at least $1 - 2\delta$, under the clean event.

**Bound on** ② The second term is bounded similarly to the second part of Theorem 4.3. Precisely, we notice that under the clean event $\mathcal{E}^{G,\Delta}(\delta)$, the optimal safe solution $q^*$ is included in the constrained decision space for each $t \in [T]$. Moreover we notice that, for each $t \in [T]$, the constrained space is convex and linear, by construction of the convex program. Thus, following the standard analysis of online mirror descent (Orabona, 2019) and from Lemma G.6, we have, under the clean event:

$$\text{②} \leq \frac{L \ln\left(|X|^2|A|\right)}{\eta} + \eta \sum_{t,x,a} q^{P_t, \widehat{\pi}_t}(x,a) \widehat{\ell}_t(x,a)^2.$$

Guaranteeing the safety property makes bounding the biased estimator more complex with respect to Theorem 4.3. Thus, noticing that $\lambda_t \leq \max_{i \in [m]} \left\{ \frac{L - \alpha_i}{L - \beta_i} \right\}$ and by definition of $\pi_t$, we proceed as follows:

$$\eta \sum_{t,x,a} q^{P_t, \widehat{\pi}_t}(x,a) \widehat{\ell}_t(x,a)^2 \leq \max_{i \in [m]} \left\{ \frac{L}{\alpha_i - \beta_i} \right\} \eta \sum_{t,x,a} (1 - \lambda_{t-1}) q^{P_t, \widehat{\pi}_t}(x,a) \widehat{\ell}_t(x,a)^2$$

$$\leq \max_{i \in [m]} \left\{ \frac{L}{\alpha_i - \beta_i} \right\} \eta \sum_{t,x,a} \left( q^{P_t, \pi_t}(x,a) - \lambda_{t-1} q^{P_t, \pi^\diamond}(x,a) \right) \widehat{\ell}_t(x,a)^2$$

$$\leq \max_{i \in [m]} \left\{ \frac{L}{\alpha_i - \beta_i} \right\} \eta \sum_{t,x,a} q^{P_t, \pi_t}(x,a) \widehat{\ell}_t(x,a)^2$$

The previous result is intuitive. Paying an additional $\max_{i \in [m]} \left\{ \frac{L}{\alpha_i - \beta_i} \right\}$ factor allows to relate the loss estimator $\widehat{\ell}_t$ with the policy that guides the exploration, namely, $\pi_t$. Thus, following the same steps as Theorem 4.3 we obtain, with probability $1 - \delta$, under the clean event:

$$\text{②} \leq \frac{L \ln\left(|X|^2|A|\right)}{\eta} + \max_{i \in [m]} \left\{ \frac{L}{\alpha_i - \beta_i} \right\} \eta |X||A|T + \max_{i \in [m]} \left\{ \frac{L}{\alpha_i - \beta_i} \right\} \frac{\eta L \ln(L/\delta)}{2\gamma}.$$

Setting $\eta = \gamma = \sqrt{\frac{L \ln(L|X||A|/\delta)}{T|X||A|}}$, we obtain:

$$\text{②} \leq \mathcal{O}\left( \max_{i \in [m]} \left\{ \frac{1}{\alpha_i - \beta_i} \right\} L \sqrt{L|X||A|T \ln\left(\frac{|X|^2|A|}{\delta}\right)} \right), \tag{20}$$

with probability at least $1 - \delta$, under the clean event.

**Bound on** ③ In the following, we show how to rewrite the third term so that the dependence on the convex combination parameter is explicit. Intuitively, the third term is the regret payed to guarantee the safety property. Thus, we rewrite the third term as follows:

$$
\begin{aligned}
\sum_{t=1}^{T} \ell_t^\top \left( q^{P_t, \pi_t} - q^{P_t, \widehat{\pi}_t} \right) &= \sum_{t=1}^{T} \ell_t^\top \left( \lambda_{t-1} q^{P_t, \pi^\diamond} + (1 - \lambda_{t-1}) q^{P_t, \widehat{\pi}_t} - q^{P_t, \widehat{\pi}_t} \right) \\
&\leq \sum_{t=1}^{T} \lambda_{t-1} \ell_t^\top q^{P_t, \pi^\diamond} \\
&\leq L \sum_{t=1}^{T} \lambda_{t-1}
\end{aligned}
$$

where we used that $\ell_t^\top q^{P_t, \pi^\diamond} \leq L$ for any $t \in [T]$. Thus, we proceed bounding $\sum_{t=1}^{T} \lambda_{t-1}$.

We focus on a single episode $t \in [T]$, in which we assume without loss of generality that the $i$-th constraint is the hardest to satisfy.

Precisely,

$$
\begin{aligned}
\lambda_t &= \frac{\min \left\{ (\widehat{g}_{t,i} + \xi_t)^\top \widehat{u}_{t+1}, L \right\} - \alpha_i}{\min \left\{ (\widehat{g}_{t,i} + \xi_t)^\top \widehat{u}_{t+1}, L \right\} - \beta_i} \\
&\leq \frac{(\widehat{g}_{t,i} + \xi_t)^\top \widehat{u}_{t+1} - \alpha_i}{(\widehat{g}_{t,i} + \xi_t)^\top \widehat{u}_{t+1} - \beta_i} \\
&\leq \frac{(\widehat{g}_{t,i} + \xi_t)^\top \widehat{u}_{t+1} - \alpha_i}{\alpha_i - \beta_i} \\
&= \frac{(\widehat{g}_{t,i} - \xi_t)^\top \widehat{u}_{t+1} + 2\xi_t^\top \widehat{u}_{t+1} - \alpha_i}{\alpha_i - \beta_i} \\
&= \frac{(\widehat{g}_{t,i} - \xi_t)^\top \widehat{q}_{t+1} + (\widehat{g}_{t,i} - \xi_t)^\top (\widehat{u}_{t+1} - \widehat{q}_{t+1}) + 2\xi_t^\top \widehat{u}_{t+1} - \alpha_i}{\alpha_i - \beta_i} \\
&\leq \frac{(\widehat{g}_{t,i} - \xi_t)^\top \widehat{q}_{t+1} + \widehat{g}_{t,i}^\top (\widehat{u}_{t+1} - \widehat{q}_{t+1}) + 2\xi_t^\top \widehat{u}_{t+1} - \alpha_i}{\alpha_i - \beta_i} \\
&\leq \frac{\widehat{g}_{t,i}^\top (\widehat{u}_{t+1} - \widehat{q}_{t+1}) + 2\xi_t^\top \widehat{u}_{t+1}}{\alpha_i - \beta_i} \\
&= \frac{\widehat{g}_{t,i}^\top (\widehat{u}_{t+1} - q^{P, \widehat{\pi}_{t+1}}) + \widehat{g}_{t,i}^\top (q^{P, \widehat{\pi}_{t+1}} - q^{P_{t+1}, \widehat{\pi}_{t+1}}) + 2\xi_t^\top \widehat{u}_{t+1}}{\alpha_i - \beta_i} \\
&\leq \frac{\|\widehat{g}_{t,i}\|_\infty \|\widehat{u}_{t+1} - q^{P, \widehat{\pi}_{t+1}}\|_1 + \|\widehat{g}_{t,i}\|_\infty \|q^{P, \widehat{\pi}_{t+1}} - q^{P_{t+1}, \widehat{\pi}_{t+1}}\|_1 + 2\xi_t^\top \widehat{u}_{t+1}}{\alpha_i - \beta_i} \\
&\leq \frac{\|\widehat{u}_{t+1} - q^{P, \widehat{\pi}_{t+1}}\|_1 + \|q^{P, \widehat{\pi}_{t+1}} - q^{P_{t+1}, \widehat{\pi}_{t+1}}\|_1 + 2\xi_t^\top \widehat{u}_{t+1}}{\alpha_i - \beta_i} \\
&\leq \frac{L(1 - \lambda_t)\|\widehat{u}_{t+1} - q^{P, \widehat{\pi}_{t+1}}\|_1 + L(1 - \lambda_t)\|q^{P, \widehat{\pi}_{t+1}} - q^{P_{t+1}, \widehat{\pi}_{t+1}}\|_1 + 2L(1 - \lambda_t)\xi_t^\top \widehat{u}_{t+1}}{\min \left\{ (\alpha_i - \beta_i), (\alpha_i - \beta_i)^2 \right\}}
\end{aligned}
$$

(21) (22) (23)

where Inequality (21) holds since, for the hardest constraint, when $\lambda_t \neq 0$, $(\widehat{g}_{t,i} + \xi_t)^\top \widehat{u}_{t+1} > \alpha_i$, Inequality (22) holds since, under the clean event, $(\widehat{g}_{t,i} - \xi_t)^\top \widehat{q}_{t+1} \leq \alpha_i$ and Inequality (23) holds since $\lambda_t \leq \frac{L - \alpha_i}{L - \beta_i}$. Intuitively, Inequality (23) shows that, to guarantee the safety property, Algorithm 3 has to pay a factor proportional to the pessimism introduced on the transition and cost functions, plus the constraints satisfaction gap of the strictly feasible solution given as input to the algorithm.

We need to generalize the result summing over $t$, taking into account that the hardest constraints may vary. Thus, we bound

the summation as follows,

$$\sum_{t=1}^{T} \lambda_{t-1} \leq \max_{i \in [m]} \left\{ \frac{2L}{\min\left\{(\alpha_i - \beta_i), (\alpha_i - \beta_i)^2\right\}} \right\} \cdot$$

$$\cdot \sum_{t=1}^{T} \left( (1 - \lambda_{t-1}) \left( \|\widehat{u}_t - q^{P,\widehat{\pi}_t}\|_1 + \|q^{P,\widehat{\pi}_t} - q^{P_t,\widehat{\pi}_t}\|_1 + \xi_{t-1}^\top \widehat{u}_t \right) \right)$$

The first two terms of the equation are bounded applying Lemma D.1, which holds with probability at least $1 - 2\delta$, under the clean event, while, to bound $\sum_{t=1}^{T} (1 - \lambda_{t-1}) \xi_{t-1}^\top \widehat{u}_t$, we proceed as follows:

$$\sum_{t=1}^{T} (1 - \lambda_{t-1}) \xi_{t-1}^\top \widehat{u}_t = \sum_{t=1}^{T} (1 - \lambda_{t-1}) \xi_{t-1}^\top q^{P,\widehat{\pi}_t} + \sum_{t=1}^{T} (1 - \lambda_{t-1}) \xi_{t-1}^\top (\widehat{u}_t - q^{P,\widehat{\pi}_t}),$$

where the second term is bounded employing Hölder inequality and Lemma D.1. Next, we focus on the first term, proceeding as follows,

$$\sum_{t=1}^{T} (1 - \lambda_{t-1}) \xi_{t-1}^\top q^{P,\widehat{\pi}_t}$$

$$\leq \sum_{t=1}^{T} \xi_{t-1}^\top q_t \tag{24}$$

$$\leq \sum_{t=1}^{T} \sum_{x,a} \xi_{t-1}(x,a) \mathbb{1}_t(x,a) + L\sqrt{2T \ln \frac{1}{\delta}} \tag{25}$$

$$= \sqrt{4 \ln\left(\frac{T|X||A|m}{\delta}\right)} \sum_{t=1}^{T} \sum_{x,a} \sqrt{\frac{1}{\max\{1, N_{t-1}(x,a)\}}} \mathbb{1}_t(x,a) + L\sqrt{2T \ln \frac{1}{\delta}}$$

$$\leq 6\sqrt{\ln\left(\frac{T|X||A|m}{\delta}\right)} \sqrt{|X||A| \sum_{x,a} N_T(x,a)} + L\sqrt{2T \ln \frac{1}{\delta}} \tag{26}$$

$$\leq 6\sqrt{L|X||A|T \ln\left(\frac{T|X||A|m}{\delta}\right)} + L\sqrt{2T \ln \frac{1}{\delta}},$$

where Inequality (24) follows from the definition of $\pi_t$, Inequality (25) follows from Azuma-Hoeffding inequality and Inequality (26) holds since $1 + \sum_{t=1}^{T} \frac{1}{\sqrt{t}} \leq 2\sqrt{T} + 1 \leq 3\sqrt{T}$ and Cauchy-Schwarz inequality.

Thus, we obtain,

$$③ \leq \mathcal{O}\left( \max_{i \in [m]} \left\{ \frac{1}{\min\left\{(\alpha_i - \beta_i), (\alpha_i - \beta_i)^2\right\}} \right\} L^3 |X| \sqrt{|A|T \ln\left(\frac{T|X||A|m}{\delta}\right)} \right), \tag{27}$$

with probability at least $1 - 3\delta$, under the clean event.

**Bound on ④** We first notice that ④ presents an additional challenge with respect to the bounded violation case. Indeed, since $\widehat{\pi}_t$ is not the policy that drives the exploration, $\widehat{\ell}_t$ cannot be directly bounded employing results from the unconstrained adversarial MDPs literature. First, we rewrite the fourth term as follows,

$$\sum_{t=1}^{T} \left( \ell_t - \widehat{\ell}_t \right)^\top q^{P_t,\widehat{\pi}_t} \leq \sum_{t=1}^{T} \left( \mathbb{E}_t[\widehat{\ell}_t] - \widehat{\ell}_t \right)^\top q^{P_t,\widehat{\pi}_t} + \sum_{t=1}^{T} \left( \ell_t - \mathbb{E}_t[\widehat{\ell}_t] \right)^\top q^{P_t,\widehat{\pi}_t},$$

where $\mathbb{E}_t[\cdot]$ is the expectation given the filtration up to time $t$. To bound the first term we employ the Azuma-Hoeffding inequality noticing that, the martingale difference sequence is bounded by:

$$\widehat{\ell}_t^\top q^{P_t,\widehat{\pi}_t} \leq \max_{i \in [m]} \left\{ \frac{L}{\alpha_i - \beta_i} \right\} \widehat{\ell}_t^\top (1 - \lambda_{t-1}) q^{P_t,\widehat{\pi}_t}$$

$$= \max_{i \in [m]} \left\{ \frac{L}{\alpha_i - \beta_i} \right\} \widehat{\ell}_t^\top \left( q^{P_t, \pi_t} - \lambda_{t-1} q^{P_t, \pi^\diamond} \right)$$

$$\leq \max_{i \in [m]} \left\{ \frac{L}{\alpha_i - \beta_i} \right\} \widehat{\ell}_t^\top q^{P_t, \pi_t}$$

$$\leq \max_{i \in [m]} \left\{ \frac{L}{\alpha_i - \beta_i} \right\} L,$$

where the first inequality holds since $\lambda_{t-1} \leq \lambda_0$. Thus, the first term is bounded by $\max_{i \in [m]} \left\{ \frac{L}{\alpha_i - \beta_i} \right\} L \sqrt{2T \ln \frac{1}{\delta}}$. To bound the second term, we employ the definition of $\pi_t$ and the upper-bound to $\lambda_{t-1}$, proceeding as follows:

$$\sum_{t=1}^T \left( \ell_t - \mathbb{E}_t[\widehat{\ell}_t] \right)^\top q^{P_t, \widehat{\pi}_t}$$

$$= \sum_{t,x,a} q^{P_t, \widehat{\pi}_t}(x,a) \ell_t(x,a) \left( 1 - \frac{\mathbb{E}_t \left[ \mathbb{1}_t(x,a) \right]}{u_t(x,a) + \gamma} \right)$$

$$= \sum_{t,x,a} q^{P_t, \widehat{\pi}_t}(x,a) \ell_t(x,a) \left( 1 - \frac{q_t(x,a)}{u_t(x,a) + \gamma} \right)$$

$$\leq \max_{i \in [m]} \left\{ \frac{L}{\alpha_i - \beta_i} \right\} \sum_{t,x,a} (1 - \lambda_{t-1}) q^{P_t, \widehat{\pi}_t}(x,a) \ell_t(x,a) \left( 1 - \frac{q_t(x,a)}{u_t(x,a) + \gamma} \right)$$

$$\leq \max_{i \in [m]} \left\{ \frac{L}{\alpha_i - \beta_i} \right\} \sum_{t,x,a} q^{P_t, \pi_t}(x,a) \ell_t(x,a) \left( 1 - \frac{q_t(x,a)}{u_t(x,a) + \gamma} \right)$$

$$= \max_{i \in [m]} \left\{ \frac{L}{\alpha_i - \beta_i} \right\} \sum_{t,x,a} \frac{q^{P_t, \pi_t}(x,a)}{u_t(x,a) + \gamma} \left( u_t(x,a) - q_t(x,a) + \gamma \right)$$

$$\leq \mathcal{O} \left( \max_{i \in [m]} \left\{ \frac{L}{\alpha_i - \beta_i} \right\} L|X| \sqrt{|A|T \ln \left( \frac{T|X||A|}{\delta} \right)} \right) + \max_{i \in [m]} \left\{ \frac{L}{\alpha_i - \beta_i} \right\} \gamma |X||A|T,$$

where the last steps holds by Lemma G.2. Thus, combining the previous equations, we have, with probability at least $1 - 3\delta$, under the clean event:

$$④ \leq \mathcal{O} \left( \max_{i \in [m]} \left\{ \frac{1}{\alpha_i - \beta_i} \right\} L^2 |X| \sqrt{|A|T \ln \left( \frac{T|X||A|}{\delta} \right)} \right) \tag{28}$$

**Bound on ⑤** The last term is bounded as in Theorem 4.3. Thus, setting $\gamma = \sqrt{\frac{L \ln(L|X||A|/\delta)}{T|X||A|}}$, we obtain, with probability at least $1 - \delta$, under the clean event:

$$⑤ \leq \mathcal{O} \left( L \sqrt{|X||A|T \ln \left( \frac{T|X||A|}{\delta} \right)} \right). \tag{29}$$

**Final result** Finally, we combine the bounds on ①, ②, ③, ④ and ⑤. Applying a union bound, we obtain, with probability at least $1 - 11\delta$,

$$R_T \leq \mathcal{O} \left( \max_{i \in [m]} \left\{ \frac{1}{\min \left\{ (\alpha_i - \beta_i), (\alpha_i - \beta_i)^2 \right\}} \right\} L^3 |X| \sqrt{|A|T \ln \left( \frac{T|X||A|m}{\delta} \right)} \right),$$

which concludes the proof. $\qquad\square$

## E. Omitted proofs for constant violation

In this section, we show how it is possible to relax Condition 2.5 and still achieving constant cumulative positive violation.

### E.1. Slater's parameter estimation

We start by showing that the number of episodes necessary to Algorithm 4 to estimate the Slater's parameter are upper bounded by a constant term. This is done by means of the following lemma.

**Lemma 6.1.** *Given any $\delta \in (0,1)$, the episodes that Algorithm 4 uses to compute $\widehat{\rho}$ and $\widehat{\pi}^\diamond$ are $\bar{t} \leq 1/\rho^4 (3C_\mathcal{A}^P + 10L \ln \frac{1}{\delta} + 3C_\mathcal{A}^D + L)^4$, with prob. at least $1 - (C_P^\delta + 2)\delta$.*

*Proof.* By the no-regret property of Algorithm $\mathcal{A}^P$, it holds:

$$\sum_{t=1}^{\bar{t}} \sum_{i \in [m]} \phi_{t,i} \left( g_{t,i}^\top (q_t - q^\diamond) - \alpha_i \right) \leq C_\mathcal{A}^P \sqrt{\bar{t} \ln(\bar{t})},$$

with probability at least $1 - C_P^\delta \delta$.

Thus, applying the Azuma inequality, we get:

$$\sum_{t=1}^{\bar{t}} \sum_{i \in [m]} \phi_{t,i} (g_{t,i} - \alpha_i/L)^\top q_t \leq C_\mathcal{A}^P \sqrt{\bar{t} \ln(\bar{t})} + L\sqrt{2\bar{t} \ln \frac{1}{\delta}} + \sum_{t=1}^{\bar{t}} \sum_{i \in [m]} \phi_{t,i} (\bar{g}_i - \alpha_i/L)^\top q^\diamond$$

$$= C_\mathcal{A}^P \sqrt{\bar{t} \ln(\bar{t})} + L\sqrt{2\bar{t} \ln \frac{1}{\delta}} - \sum_{t=1}^{\bar{t}} \sum_{i \in [m]} \phi_{t,i} (\alpha_i - \beta_i)$$

$$\leq C_\mathcal{A}^P \sqrt{\bar{t} \ln(\bar{t})} + L\sqrt{2\bar{t} \ln \frac{1}{\delta}} - \bar{t} \min_{i \in [m]} (\alpha_i - \beta_i)$$

$$= C_\mathcal{A}^P \sqrt{\bar{t} \ln(\bar{t})} + L\sqrt{2\bar{t} \ln \frac{1}{\delta}} - \bar{t}\rho,$$

with probability $1 - (C_P^\delta + 1)\delta$, by Union Bound. Similarly, applying the Azuma inequality, it holds:

$$\sum_{t=1}^{\bar{t}_i} \sum_{i \in [m]} \phi_{t,i} \sum_{k=1}^{L-1} (g_{t,i}(x_k, a_k) - \alpha_i/L) \leq C_\mathcal{A}^P \sqrt{\bar{t} \ln(\bar{t})} + 2L\sqrt{2\bar{t} \ln \frac{1}{\delta}} - \bar{t}\rho,$$

with probability at least $1 - (C_P^\delta + 2)\delta$.

By the no-regret property of Algorithm $\mathcal{A}^D$, it holds:

$$-\sum_{t=1}^{\bar{t}} \sum_{i \in [m]} \phi_{t,i} \left( \sum_{k=1}^{L-1} g_{t,i}(x_k, a_k) - \alpha_i \right) - \sum_{t=1}^{\bar{t}} - \left( \sum_{k=1}^{L-1} g_{t,i^*}(x_k, a_k) - \alpha_{i^*} \right) \leq C_\mathcal{A}^D \sqrt{\bar{t}},$$

where $i^* = \arg\max_{i \in [m]} \sum_{t=1}^{\bar{t}} \sum_{k=1}^{L-1} (g_{t,i}(x_k, a_k) - \alpha_i/L)$, from which we obtain, with probability at least $1 - (C_P^\delta + 2)\delta$:

$$\max_{i \in [m]} \sum_{t=1}^{\bar{t}} \left( \sum_{k=1}^{L-1} g_{t,i}(x_k, a_k) - \alpha_i \right) \leq C_\mathcal{A}^P \sqrt{\bar{t} \ln(\bar{t})} + 2L\sqrt{2\bar{t} \ln \frac{1}{\delta}} + C_\mathcal{A}^D \sqrt{\bar{t}} - \bar{t}\rho.$$

By the stopping condition of Algorithm 4, it holds $-\max_{i \in [m]} \sum_{t=1}^{\bar{t}} \sum_{k=1}^{L-1} (g_{t,i}(x_k, a_k) - \alpha_i/L) \geq 2C_\mathcal{A}^P \sqrt{\bar{t} \ln(\bar{t})} + 8L\sqrt{2\bar{t} \ln \frac{1}{\delta}} + 2C_\mathcal{A}^D \sqrt{\bar{t}}$, which implies:

$$-\max_{i \in [m]} \sum_{t=1}^{\bar{t}-1} \sum_{k=1}^{L-1} (g_{t,i}(x_k, a_k) - \alpha_i/L) \leq 2C_\mathcal{A}^P \sqrt{\bar{t}-1 \ln(\bar{t}-1)} + 8L\sqrt{2\bar{t}-1 \ln \frac{1}{\delta}} + 2C_\mathcal{A}^D \sqrt{\bar{t}-1}$$

$$\leq 2C_\mathcal{A}^P \sqrt{\bar{t} \ln(\bar{t})} + 8L\sqrt{2\bar{t} \ln \frac{1}{\delta}} + 2C_\mathcal{A}^D \sqrt{\bar{t}},$$

and

$$\max_{i \in [m]} \sum_{t=1}^{\bar{t}-1} \sum_{k=1}^{L-1} (g_{t,i}(x_k, a_k) - \alpha_i/L) \geq -2C_{\mathcal{A}}^P \sqrt{\bar{t}\ln(\bar{t})} - 8L\sqrt{2\bar{t}\ln\frac{1}{\delta}} - 2C_{\mathcal{A}}^D \sqrt{\bar{t}}.$$

Thus, we get the following inequality:

$$-2C_{\mathcal{A}}^P \sqrt{\bar{t}\ln(\bar{t})} - 8L\sqrt{2\bar{t}\ln\frac{1}{\delta}} - 2C_{\mathcal{A}}^D \sqrt{\bar{t}} - L \leq C_{\mathcal{A}}^P \sqrt{\bar{t}\ln(\bar{t})} + 2L\sqrt{2\bar{t}\ln\frac{1}{\delta}} + C_{\mathcal{A}}^D \sqrt{\bar{t}} - \bar{t}\rho.$$

Hence,

$$\bar{t}\rho \leq 3C_{\mathcal{A}}^P \sqrt{\bar{t}\ln(\bar{t})} + 10L\sqrt{2\bar{t}\ln\frac{1}{\delta}} + 3C_{\mathcal{A}}^D \sqrt{\bar{t}} + L$$

$$\leq 3C_{\mathcal{A}}^P \bar{t}^{3/4} + 10L\sqrt{2\ln\frac{1}{\delta}}\bar{t}^{3/4} + 3C_{\mathcal{A}}^D \bar{t}^{3/4} + L\bar{t}^{3/4},$$

which implies:

$$\bar{t} \leq \frac{\left(3C_{\mathcal{A}}^P + 10L\sqrt{2\ln\frac{1}{\delta}} + 3C_{\mathcal{A}}^D + L\right)^4}{\rho^4}.$$

This concludes the proof. $\qquad\square$

Thus, we show that the estimation of the Slater's parameter $\widehat{\rho}$ is upper bounded by the true one. A similar reasoning holds for the estimated strictly feasible solution. The result is provided in the following lemma.

**Lemma 6.2.** *Given any $\delta \in (0,1)$, Algorithm 4 guarantees $\widehat{\rho} \leq \min_{i \in [m]}(\alpha_i - \overline{g}_i^\top q^{P,\widehat{\pi}^\diamond}) \leq \rho$ with prob. at least $1 - 2\delta$.*

*Proof.* It holds:

$$\widehat{\rho} = -\frac{1}{\bar{t}}\left(\max_{i \in [m]} \sum_{t=1}^{\bar{t}} \left(\sum_{k=1}^{L-1} g_{t,i}(x_k, a_k) - \alpha_i/L\right) + 2L\sqrt{2\bar{t}\ln\frac{1}{\delta}}\right)$$

$$= \min_{i \in [m]}\left(\alpha_i - \frac{1}{\bar{t}}\left(\sum_{t=1}^{\bar{t}}\sum_{k=1}^{L-1} g_{t,i}(x_k, a_k) + 2L\sqrt{2\bar{t}\ln\frac{1}{\delta}}\right)\right)$$

$$\leq \min_{i \in [m]}\left(\alpha_i - \frac{1}{\bar{t}}\left(\sum_{t=1}^{\bar{t}}\sum_{k=1}^{L-1} \overline{g}_i(x_k, a_k) - L\sqrt{2\bar{t}\ln\frac{1}{\delta}} + 2L\sqrt{2\bar{t}\ln\frac{1}{\delta}}\right)\right)$$

$$\leq \min_{i \in [m]}\left(\alpha_i - \frac{1}{\bar{t}}\left(\sum_{t=1}^{\bar{t}} \overline{g}_i^\top q_t - 2L\sqrt{2\bar{t}\ln\frac{1}{\delta}} + 2L\sqrt{2\bar{t}\ln\frac{1}{\delta}}\right)\right)$$

$$= \min_{i \in [m]}\left(\alpha_i + \frac{1}{\bar{t}}\left(\sum_{t=1}^{\bar{t}} -\overline{g}_i^\top q_t + 2L\sqrt{2\bar{t}\ln\frac{1}{\delta}} - 2L\sqrt{2\bar{t}\ln\frac{1}{\delta}}\right)\right)$$

$$\leq \min_{i \in [m]}(\alpha_i - \beta_i)$$

$$= \rho,$$

where the first steps hold with probability at least $1 - 2\delta$ by Azuma inequality and a Union Bound and the last inequality holds by definition of $q^\diamond$.

To prove the second result we first notice that, by definition of $q^\diamond$:

$$\min_{i \in [m]}(\alpha_i - \overline{g}_i^\top q^{P,\widehat{\pi}^\diamond}) \leq \min_{i \in [m]}(\alpha_i - \overline{g}_i^\top q^\diamond) = \min_{i \in [m]}(\alpha_i - \beta_i) = \rho.$$

Furthermore, by definition of $\widehat{\pi}^\diamond$, it holds:

$$q^{P,\widehat{\pi}^\diamond} = \frac{1}{\bar{t}} \sum_{t=1}^{\bar{t}} q^{P,\pi_t}.$$

Hence,

$$
\begin{aligned}
\min_{i \in [m]} \left( \alpha_i - \overline{g}_i^\top q^{P,\widehat{\pi}^\diamond} \right) &= \min_{i \in [m]} \left( \alpha_i - \frac{1}{\bar{t}} \left( \sum_{t=1}^{\bar{t}} \overline{g}_i^\top q^{P,\pi_t} \right) \right) \\
&\geq \min_{i \in [m]} \left( \alpha_i - \frac{1}{\bar{t}} \left( \sum_{t=1}^{\bar{t}} \sum_{k=1}^{L-1} g_{t,i}(x_k, a_k) + 2L\sqrt{2\bar{t} \ln \frac{1}{\delta}} \right) \right) \\
&= \widehat{\rho},
\end{aligned}
$$

where the first inequality holds with probability at least $1 - 2\delta$ employing the Azuma inequality and a union bound. This concludes the proof. $\qquad\square$

Finally, we show that $\widehat{\rho}$ is not to small, namely, it is lower bounded by $\rho/2$. The result is provided in the following lemma.

**Lemma 6.4.** *Given any $\delta \in (0,1)$, Algorithm 4 guarantees $\widehat{\rho} \geq \rho/2$ with probability at least $1 - (C_P^\delta + 2)\delta$.*

*Proof.* Similarly to Lemma 6.1, we get, with probability at least $1 - (C_P^\delta + 2)\delta$

$$-\max_{i \in [m]} \sum_{t=1}^{\bar{t}} \left( \sum_{k=1}^{L-1} g_{t,i}(x_k, a_k) - \alpha_i \right) \geq -C_{\mathcal{A}}^P \sqrt{\bar{t} \ln(\bar{t})} - 2L\sqrt{2\bar{t} \ln \frac{1}{\delta}} - C_{\mathcal{A}}^D \sqrt{\bar{t}} + \bar{t}\rho.$$

Thus, notice that, by the stopping condition of Algorithm 4, it holds:

$$
\begin{aligned}
&-\max_{i \in [m]} \sum_{t=1}^{\bar{t}} \left( \sum_{k=1}^{L-1} g_{t,i}(x_k, a_k) - \alpha_i \right) - C_{\mathcal{A}}^P \sqrt{\bar{t} \ln(\bar{t})} - 4L\sqrt{2\bar{t} \ln \frac{1}{\delta}} - C_{\mathcal{A}}^D \sqrt{\bar{t}} \\
&= -\frac{1}{2} \max_{i \in [m]} \sum_{t=1}^{\bar{t}} \left( \sum_{k=1}^{L-1} g_{t,i}(x_k, a_k) - \alpha_i \right) - \frac{1}{2} \max_{i \in [m]} \sum_{t=1}^{\bar{t}} \left( \sum_{k=1}^{L-1} g_{t,i}(x_k, a_k) - \alpha_i \right) - C_{\mathcal{A}}^P \sqrt{\bar{t} \ln(\bar{t})} - 4L\sqrt{2\bar{t} \ln \frac{1}{\delta}} - C_{\mathcal{A}}^D \sqrt{\bar{t}} \\
&\geq -\frac{1}{2} \max_{i \in [m]} \sum_{t=1}^{\bar{t}} \left( \sum_{k=1}^{L-1} g_{t,i}(x_k, a_k) - \alpha_i \right) + C_{\mathcal{A}}^P \sqrt{\bar{t} \ln(\bar{t})} - C_{\mathcal{A}}^P \sqrt{\bar{t} \ln(\bar{t})} + 4L\sqrt{2\bar{t} \ln \frac{1}{\delta}} - 4L\sqrt{2\bar{t} \ln \frac{1}{\delta}} + C_{\mathcal{A}}^D \sqrt{\bar{t}} - C_{\mathcal{A}}^D \sqrt{\bar{t}} \\
&\geq -\frac{1}{2} \max_{i \in [m]} \sum_{t=1}^{\bar{t}} \left( \sum_{k=1}^{L-1} g_{t,i}(x_k, a_k) - \alpha_i \right).
\end{aligned}
$$

Hence,

$$
\begin{aligned}
\widehat{\rho} &= -\frac{1}{\bar{t}} \left( \max_{i \in [m]} \sum_{t=1}^{\bar{t}} \left( \sum_{k=1}^{L-1} g_{t,i}(x_k, a_k) - \alpha_i/L \right) + 2L\sqrt{2\bar{t} \ln \frac{1}{\delta}} \right) \\
&= -\frac{1}{\bar{t}} \left( \max_{i \in [m]} \sum_{t=1}^{\bar{t}} \left( \sum_{k=1}^{L-1} g_{t,i}(x_k, a_k) - \alpha_i/L \right) \pm C_{\mathcal{A}}^P \sqrt{\bar{t} \ln(\bar{t})} \pm 4L\sqrt{2\bar{t} \ln \frac{1}{\delta}} \pm C_{\mathcal{A}}^D \sqrt{\bar{t}} + 2L\sqrt{2\bar{t} \ln \frac{1}{\delta}} \right) \\
&\geq \frac{1}{\bar{t}} \left( -\frac{1}{2} \max_{i \in [m]} \sum_{t=1}^{\bar{t}} \left( \sum_{k=1}^{L-1} g_{t,i}(x_k, a_k) - \alpha_i \right) + C_{\mathcal{A}}^P \sqrt{\bar{t} \ln(\bar{t})} + 4L\sqrt{2\bar{t} \ln \frac{1}{\delta}} + C_{\mathcal{A}}^D \sqrt{\bar{t}} - 2L\sqrt{2\bar{t} \ln \frac{1}{\delta}} \right) \\
&\geq \frac{\rho}{2}.
\end{aligned}
$$

This concludes the proof. $\qquad\square$

## E.2. Violations

In this section, we provide the theoretical guarantees attained by Algorithm 4 in terms of cumulative positive violation. This is done by means of the following theorem.

**Theorem 6.3.** *Given $\delta \in (0, 1)$, Algorithm 4 attains $V_T \leq \mathcal{O}(L/\rho^4(C_{\mathcal{A}}^P + L\sqrt{\ln \frac{1}{\delta}} + C_{\mathcal{A}}^D + L)^4)$ with probability at least $1 - (C_P^\delta + 7)\delta$.*

*Proof.* We split the violations between the two phases of Algorithm 4 as:

$$
\begin{aligned}
V_T &\leq \max_{i \in [m]} \sum_{t=1}^{\bar{t}} \left[\bar{g}_i^\top q_t - \alpha_i\right]^+ + \max_{i \in [m]} \sum_{t=\bar{t}+1}^{T} \left[\bar{g}_i^\top q_t - \alpha_i\right]^+ \\
&\leq L\bar{t} + \max_{i \in [m]} \sum_{t=\bar{t}}^{T} \left[\bar{g}_i^\top q_t - \alpha_i\right]^+ \\
&\leq \mathcal{O}\left(\frac{L\left(C_{\mathcal{A}}^P + L\sqrt{\ln \frac{1}{\delta}} + C_{\mathcal{A}}^D + L\right)^4}{\rho^4}\right) + \max_{i \in [m]} \sum_{t=\bar{t}+1}^{T} \left[\bar{g}_i^\top q_t - \alpha_i\right]^+,
\end{aligned}
$$

where the last step holds with probability at least $1 - (C_P^\delta + 2)\delta$ by Lemma 6.1.

In the following we show that, after $\bar{t}$ episodes, Algorithm 4 is safe with high probability. Similarly to Theorem 5.1 there are two possible scenarios defined by $\lambda_t$, namely, $\lambda_t = 0$ and $\lambda_t \in (0, 1)$. When $\lambda_t = 0$, applying the same reasoning of Theorem 5.1 gives the result.

In the following analysis we consider a generic constraints $i \in [m]$. Thus, we notice that the constraints cost attained by the *non-Markovian* policy $\pi_t$, is equal to $\lambda_{t-1}\bar{g}_i^\top q^{P,\hat{\pi}^\diamond} + (1 - \lambda_{t-1})\bar{g}_i^\top q^{P,\hat{\pi}_t}$.

Then, we consider both the cases when $L < (\hat{g}_{t-1,i} + \xi_{t-1})^\top \hat{u}_t$ (first case) and $L > (\hat{g}_{t-1,i} + \xi_{t-1})^\top \hat{u}_t$ (second case). If the two quantities are equivalent, the proof still holds breaking the ties arbitrarily.

*First case.* It holds that:

$$
\begin{aligned}
\lambda_{t-1}\bar{g}_i^\top q^{P,\hat{\pi}^\diamond} + (1 - \lambda_{t-1})\bar{g}_i^\top q^{\hat{\pi}_t,P} &= \frac{L - \alpha_i}{L - \alpha_i + \hat{\rho}}(\bar{g}_i^\top q^{P,\hat{\pi}^\diamond} - L) + L \\
&\leq \frac{\alpha_i - L}{\alpha_i - \hat{\rho} - L}(\alpha_i - \hat{\rho} - L) + L \qquad (30) \\
&= \alpha_i,
\end{aligned}
$$

where Inequality (30) holds with probability at least $1 - 2\delta$ thanks to Lemma 6.2.

*Second case.* Similarly to the first case, it holds that, under the clean event:

$$
\begin{aligned}
\lambda_{t-1}\bar{g}_i^\top q^{P,\hat{\pi}^\diamond} &+ (1 - \lambda_{t-1})\bar{g}_i^\top q^{P,\hat{\pi}_t} \\
&\leq \lambda_{t-1}\bar{g}_i^\top q^{P,\hat{\pi}^\diamond} + (1 - \lambda_{t-1})\left(\hat{g}_{t-1,i} + \xi_{t-1}\right)^\top q^{P,\hat{\pi}_t} \qquad (31) \\
&\leq \lambda_{t-1}\bar{g}_i^\top q^{P,\hat{\pi}^\diamond} + (1 - \lambda_{t-1})\left(\hat{g}_{t-1,i} + \xi_{t-1}\right)^\top \hat{u}_t \qquad (32) \\
&= \lambda_{t-1}\bar{g}_i^\top q^{P,\hat{\pi}^\diamond} - \lambda_{t-1}\left(\hat{g}_{t-1,i} + \xi_{t-1}\right)^\top \hat{u}_t + \left(\hat{g}_{t-1,i} + \xi_{t-1}\right)^\top \hat{u}_t \\
&= \lambda_{t-1}(\bar{g}_i^\top q^{P,\hat{\pi}^\diamond} - \left(\hat{g}_{t-1,i} + \xi_{t-1}\right)^\top \hat{u}_t) + \left(\hat{g}_{t-1,i} + \xi_{t-1}\right)^\top \hat{u}_t \\
&\leq \frac{\left(\hat{g}_{t-1,i} + \xi_{t-1}\right)^\top \hat{u}_t - \alpha_i}{\left(\hat{g}_{t-1,i} + \xi_{t-1}\right)^\top \hat{u}_t - \alpha_i + \hat{\rho}}(\alpha_i - \hat{\rho} - \left(\hat{g}_{t-1,i} + \xi_{t-1}\right)^\top \hat{u}_t) + \left(\hat{g}_{t-1,i} + \xi_{t-1}\right)^\top \hat{u}_t \\
&= \frac{\alpha_i - \left(\hat{g}_{t-1,i} + \xi_{t-1}\right)^\top \hat{u}_t}{\alpha_i - \hat{\rho} - \left(\hat{g}_{t-1,i} + \xi_{t-1}\right)^\top \hat{u}_t}(\alpha_i - \hat{\rho} - \left(\hat{g}_{t-1,i} + \xi_{t-1}\right)^\top \hat{u}_t) + \left(\hat{g}_{t-1,i} + \xi_{t-1}\right)^\top \hat{u}_t \\
&= \alpha_i - \left(\hat{g}_{t-1,i} + \xi_{t-1}\right)^\top \hat{u}_t + \left(\hat{g}_{t-1,i} + \xi_{t-1}\right)^\top \hat{u}_t
\end{aligned}
$$

$$= \alpha_i,$$

where Inequality (31) holds by the definition of the event and Inequality (32) holds by the definition of $\widehat{u}_t$.

To conclude the proof, we underline that $\lambda_t$ is chosen taking the maximum over the constraints, which implies that the more conservative $\lambda_t$ (the one which takes the combination nearer to the strictly feasible solution) is chosen. Thus, all the constraints are satisfied and a final union bound concludes the proof. $\qquad\square$

### E.3. Regret

In this section, we provide the theoretical guarantees attained by Algorithm 4 in terms of cumulative regret. This is done by means of the following theorem.

**Theorem 6.5.** *Given any $\delta \in (0,1)$, with $\eta = \gamma = \sqrt{L \ln(L|X||A|/\delta)/T|X||A|}$, Algorithm 4 attains regret $R_T \le \mathcal{O}(\Theta L^3 |X|\sqrt{|A|T \ln(T|X||A|m/\delta)} + L/\rho^4(C_{\mathcal{A}}^P + L \ln 1/\delta + C_{\mathcal{A}}^D + L)^4)$ with probability at least $1 - (C_P^\delta + 13)\delta$, where we let $\Theta := 1/\min\{\rho,\rho^2\}$.*

*Proof.* We split the regret between the two phases of Algorithm 4 as:

$$R_T \le \sum_{t=1}^{\bar{t}} \ell_t^\top (q^{P,\pi_t} - q^*) + \sum_{t=\bar{t}+1}^{T} \ell_t^\top (q^{P,\pi_t} - q^*)$$

$$\le L\bar{t} + \sum_{t=\bar{t}+1}^{T} \ell_t^\top (q^{P,\pi_t} - q^*)$$

$$\le \mathcal{O}\left( \frac{L \left( C_{\mathcal{A}}^P + L\sqrt{\ln \frac{1}{\delta}} + C_{\mathcal{A}}^D + L \right)^4}{\rho^4} \right) + \sum_{t=\bar{t}+1}^{T} \ell_t^\top (q^{P,\pi_t} - q^*),$$

where the last step holds with probability at least $1 - (C_P^\delta + 2)\delta$ by Lemma 6.1.

To bound the second terms we follow the steps of Theorem 5.2 after noticing that, for all $t \in [T]$:

$$\lambda_t \le \max_{i\in[m]} \left\{ \frac{L - \alpha_i}{L - \alpha_i + \widehat{\rho}} \right\} < 1,$$

and that:

$$\lambda_t = \frac{\min\left\{ (\widehat{g}_{t,i} + \xi_t)^\top \widehat{u}_{t+1}, L \right\} - \alpha_i}{\min\left\{ (\widehat{g}_{t,i} + \xi_t)^\top \widehat{u}_{t+1}, L \right\} - \alpha_i + \widehat{\rho}}$$

$$\le \frac{(\widehat{g}_{t,i} + \xi_t)^\top \widehat{u}_{t+1} - \alpha_i}{(\widehat{g}_{t,i} + \xi_t)^\top \widehat{u}_{t+1} - \alpha_i + \widehat{\rho}}$$

$$\le \frac{(\widehat{g}_{t,i} + \xi_t)^\top \widehat{u}_{t+1} - \alpha_i}{\widehat{\rho}} \tag{33}$$

$$= \frac{(\widehat{g}_{t,i} - \xi_t)^\top \widehat{u}_{t+1} + 2\xi_t^\top \widehat{u}_{t+1} - \alpha_i}{\widehat{\rho}}$$

$$= \frac{(\widehat{g}_{t,i} - \xi_t)^\top \widehat{q}_{t+1} + (\widehat{g}_{t,i} - \xi_t)^\top (\widehat{u}_{t+1} - \widehat{q}_{t+1}) + 2\xi_t^\top \widehat{u}_{t+1} - \alpha_i}{\widehat{\rho}}$$

$$\le \frac{(\widehat{g}_{t,i} - \xi_t)^\top \widehat{q}_{t+1} + \widehat{g}_{t,i}^\top (\widehat{u}_{t+1} - \widehat{q}_{t+1}) + 2\xi_t^\top \widehat{u}_{t+1} - \alpha_i}{\widehat{\rho}}$$

$$\le \frac{\widehat{g}_{t,i}^\top (\widehat{u}_{t+1} - \widehat{q}_{t+1}) + 2\xi_t^\top \widehat{u}_{t+1}}{\widehat{\rho}} \tag{34}$$

$$= \frac{\widehat{g}_{t,i}^\top (\widehat{u}_{t+1} - q^{P,\widehat{\pi}_{t+1}}) + \widehat{g}_{t,i}^\top (q^{P,\widehat{\pi}_{t+1}} - q^{P_{t+1},\widehat{\pi}_{t+1}}) + 2\xi_t^\top \widehat{u}_{t+1}}{\widehat{\rho}}$$

$$\leq \frac{\|\widehat{g}_{t,i}\|_{\infty}\|\widehat{u}_{t+1} - q^{P,\widehat{\pi}_{t+1}}\|_1 + \|\widehat{g}_{t,i}\|_{\infty}\|q^{P,\widehat{\pi}_{t+1}} - q^{P_{t+1},\widehat{\pi}_{t+1}}\|_1 + 2\xi_t^{\top}\widehat{u}_{t+1}}{\widehat{\rho}}$$

$$\leq \frac{\|\widehat{u}_{t+1} - q^{P,\widehat{\pi}_{t+1}}\|_1 + \|q^{P,\widehat{\pi}_{t+1}} - q^{P_{t+1},\widehat{\pi}_{t+1}}\|_1 + 2\xi_t^{\top}\widehat{u}_{t+1}}{\widehat{\rho}}$$

$$\leq \frac{L(1-\lambda_t)\|\widehat{u}_{t+1} - q^{P,\widehat{\pi}_{t+1}}\|_1 + L(1-\lambda_t)\|q^{P,\widehat{\pi}_{t+1}} - q^{P_{t+1},\widehat{\pi}_{t+1}}\|_1 + 2L(1-\lambda_t)\xi_t^{\top}\widehat{u}_{t+1}}{\min\{\widehat{\rho},\widehat{\rho}^2\}} \qquad (35)$$

$$\leq \frac{4L(1-\lambda_t)\|\widehat{u}_{t+1} - q^{P,\widehat{\pi}_{t+1}}\|_1 + 4L(1-\lambda_t)\|q^{P,\widehat{\pi}_{t+1}} - q^{P_{t+1},\widehat{\pi}_{t+1}}\|_1 + 8L(1-\lambda_t)\xi_t^{\top}\widehat{u}_{t+1}}{\min\{\rho,\rho^2\}} \qquad (36)$$

where Inequality (33) holds since, for the hardest constraint, when $\lambda_t \neq 0$, $(\widehat{g}_{t,i} + \xi_t)^{\top}\widehat{u}_{t+1} > \alpha_i$, Inequality (34) holds since, under the clean event, $(\widehat{g}_{t,i} - \xi_t)^{\top}\widehat{q}_{t+1} \leq \alpha_i$, Inequality (35) holds since $\lambda_t \leq \frac{L-\alpha_i}{L-\alpha_i+\widehat{\rho}}$ and Inequality (36) holds with probability at least $1 - (C_P^{\delta} + 2)\delta$ by Lemma 6.4. A final union bound concludes the proof. □

## F. Omitted proof of the lower bound

In this section, we provide the lower bound which holds both in the second setting, that is, when the objective is to guarantee the safety property for each episode and when the objective is to attain constant violation.

**Theorem 6.6.** *There exist two instances of CMDPs (with a single state and one constraint) such that, if in the first instance an algorithm suffers from a violation $V_T = o(\sqrt{T})$ probability at least $1 - n\delta$ for any $\delta \in (0,1)$ and $n > 0$, then, in the second instance, it must suffer from a regret $R_T = \Omega(\frac{1}{\rho}\sqrt{T})$ with probability $3/4 - n\delta$.*

*Proof.* We consider two instances defined as follows. Both of them are characterized by a CMDP with one state (which is omitted for simplicity), two actions $a_1, a_2$, one constraint and $\alpha = 1/2$. For the sake of simplicity we consider CMDP with rewards in place of losses. Notice that this is without loss of generality since any losses can be converted to an associated reward. We assume that the rewards are deterministic while the constraints are Bernoulli distributions with means defined in the following. Specifically, instance $i^1$ and instance $i^2$ are defined as:

$$i^1 := \begin{cases} \overline{r}(a_1) = \frac{1}{2}, & \overline{g}(a_1) = \frac{1}{2} + \epsilon \\ \overline{r}(a_2) = 0, & \overline{g}(a_2) = \frac{1}{2} - \rho \end{cases},$$

$$i^2 := \begin{cases} \overline{r}(a_1) = \frac{1}{2}, & \overline{g}(a_1) = \frac{1}{2} \\ \overline{r}(a_2) = 0, & \overline{g}(a_2) = \frac{1}{2} - \rho \end{cases},$$

where $\epsilon$ is a parameter to be defined later. Thus, since the algorithm must suffer $o(\sqrt{T})$ violation, for any constant $c > 0$, it holds:

$$\mathbb{P}^1\left\{q_t(a_2) \geq \frac{\epsilon}{\epsilon + \rho} - c\frac{1}{\sqrt{T}}, \quad \forall t \in [T]\right\} \geq 1 - n \cdot \delta,$$

where $q(a_2)$ is the occupancy measure associated to action $a_2$ and $\mathbb{P}^1$ is the probability measure of instance $i_1$ which encompasses the randomness of both environment and algorithm. Thus we can rewrite the inequality above as:

$$\mathbb{P}^1\left\{\sum_{t=1}^{T} q_t(a_2) \geq T\frac{\epsilon}{\epsilon + \rho} - c\sqrt{T}\right\} \geq 1 - n \cdot \delta.$$

By means of the Pinsker's inequality we can relate the probability measures $\mathbb{P}^1$ and $\mathbb{P}^2$ as follows:

$$\mathbb{P}^2\left\{\sum_{t=1}^{T} q_t(a_2) \geq T\frac{\epsilon}{\epsilon + \rho} - c\sqrt{T}\right\} \geq \mathbb{P}^1\left\{\sum_{t=1}^{T} q_t(a_2) \geq T\frac{\epsilon}{\epsilon + \rho} - c\sqrt{T}\right\} - \sqrt{\frac{1}{2}KL(i^1, i^2)},$$

where $KL(i^1, i^2)$ is the the KL-divergence between the probability measures of instance $i_1$ and $i_2$.

Noticing that by standard KL-decomposition argument, $KL(i^1, i^2) \leq \epsilon^2 T$, we have:

$$\mathbb{P}^2\left\{\sum_{t=1}^{T} q_t(a_2) \geq T\frac{\epsilon}{\epsilon + \rho} - c\sqrt{T}\right\} \geq 1 - n \cdot \delta - \epsilon\sqrt{\frac{T}{2}}.$$

We then notice that, since the rewards are deterministic, the regret of the second instance $R_T^2$ is bounded as:

$$
\begin{aligned}
R_T^2 &= \frac{1}{2} \sum_{t=1}^{T} q_t(a_2) \\
&\geq \frac{1}{2} T \frac{\epsilon}{\epsilon + \rho} - \frac{c}{2} \sqrt{T} \\
&\geq \frac{1}{4\rho} T \epsilon - c \sqrt{T} \\
&= \frac{1}{16\rho} \sqrt{2T} - c\sqrt{T} \\
&\geq \frac{1}{32\rho} \sqrt{2T},
\end{aligned}
$$

with probability $\frac{3}{4} - n \cdot \delta$, taking $\epsilon = \frac{1}{4} \sqrt{\frac{2}{T}}$, $\rho \geq \epsilon$ and $c \leq \frac{\sqrt{2}}{32}$. This concludes the proof. □

## G. Auxiliary lemmas from existing works

### G.1. Auxiliary lemmas for the transitions estimation

First, we provide the formal result on the transitions confidence set.

**Lemma G.1** (Jin et al. (2020)). *Given a confidence $\delta \in (0,1)$, with probability at least $1 - 4\delta$, it holds that the transition function $P$ belongs to $\mathcal{P}_t$ for all $t \in [T]$.*

Similarly to (Jin et al., 2020), the estimated occupancy measure space $\Delta(\mathcal{P}_t)$ is characterized as follows:

$$
\Delta(\mathcal{P}_t) := \begin{cases}
\forall k, & \sum_{x \in X_k, a \in A, x' \in X_{k+1}} q(x, a, x') = 1 \\
\forall k, \forall x, & \sum_{a \in A, x' \in X_{k+1}} q(x, a, x') = \sum_{x' \in X_{k-1}, a \in A} q(x', a, x) \\
\forall k, \forall (x, a, x'), & q(x, a, x') \leq \left[ \widehat{P}_t(x' \mid x, a) + \epsilon_t(x' \mid x, a) \right] \sum_{y \in X_{k+1}} q(x, a, y) . \\
& q(x, a, x') \geq \left[ \widehat{P}_t(x' \mid x, a) - \epsilon_t(x' \mid x, a) \right] \sum_{y \in X_{k+1}} q(x, a, y) \\
& q(x, a, x') \geq 0
\end{cases}
$$

Given the estimation of the occupancy measure space, it is possible to derive the following lemma.

**Lemma G.2.** *(Jin et al., 2020) With probability at least $1 - 6\delta$, for any collection of transition functions $\{P_t^x\}_{x \in X}$ such that $P_t^x \in \mathcal{P}_t$, we have, for all $x$,*

$$
\sum_{t=1}^{T} \sum_{x \in X, a \in A} \left| q^{P_t^x, \pi_t}(x, a) - q_t(x, a) \right| \leq \mathcal{O}\left( L|X| \sqrt{|A|T \ln \left( \frac{T|X||A|}{\delta} \right)} \right).
$$

We underline that the constrained space defined by Program (2) is a subset of $\Delta(\mathcal{P}_t)$. This implies that, in Algorithm 2, it holds $\widehat{q}_t \in \Delta(\mathcal{P}_t)$ and Lemma G.2 is valid.

### G.2. Auxiliary lemmas for the optimistic loss estimator

We will make use of the optimistic biased estimator with implicit exploration factor (see, (Neu, 2015)). Precisely, we define the loss estimator as follows, for all $t \in [T]$:

$$
\widehat{\ell}_t(x, a) := \frac{\ell_t(x, a)}{u_t(x, a) + \gamma} \mathbb{1}_t\{x, a\}, \quad \forall (x, a) \in X \times A,
$$

where $u_t(x, a) := \max_{\overline{P} \in \mathcal{P}_t} q^{\overline{P}, \pi_t}(x, a)$. Thus, the following lemmas hold.

**Lemma G.3.** *(Jin et al., 2020)* *For any sequence of functions* $\alpha_1, \ldots, \alpha_T$ *such that* $\alpha_t \in [0, 2\gamma]^{X \times A}$ *is* $\mathcal{F}_t$*-measurable for all* $t$*, we have with probability at least* $1 - \delta$*,*

$$\sum_{t=1}^{T} \sum_{x,a} \alpha_t(x, a) \left( \widehat{\ell}_t(x, a) - \frac{q_t(x, a)}{u_t(x, a)} \ell_t(x, a) \right) \leq L \ln \frac{L}{\delta}.$$

Following the analysis of Lemma G.3, with $\alpha_t(x, a) = 2\gamma \mathbb{1}_t(x, a)$ and union bound, the following corollary holds.

**Corollary G.4.** *(Jin et al., 2020)* *With probability at least* $1 - \delta$*:*

$$\sum_{t=1}^{T} \left( \widehat{\ell}_t(x, a) - \frac{q_t(x, a)}{u_t(x, a)} \ell_t(x, a) \right) \leq \frac{1}{2\gamma} \ln \left( \frac{|X||A|}{\delta} \right).$$

Furthermore, when $\pi_t \leftarrow \widehat{q}_t$, the following lemma holds.

**Lemma G.5.** *(Jin et al., 2020)* *With probability at least* $1 - 7\delta$*,*

$$\sum_{t=1}^{T} \left( \ell_t - \widehat{\ell}_t \right)^{\top} \widehat{q}_t \leq \mathcal{O} \left( L|X| \sqrt{|A|T \ln \left( \frac{T|X||A|}{\delta} \right)} + \gamma |X||A|T \right).$$

We notice that $\pi_t \leftarrow \widehat{q}_t$ holds only for Algorithm 2, since in Algorithm 3, $\pi_t \leftarrow \widehat{q}_t$ with probability $1 - \lambda_{t-1}$.

### G.3. Auxiliary lemmas for online mirror descent

We will employ the following results for OMD (see, Orabona (2019)) with uniform initialization over the estimated occupancy measure space.

**Lemma G.6.** *(Jin et al., 2020)* *The OMD update with* $\widehat{q}_1(x, a, x') = \frac{1}{|X_k||A||X_{k+1}|}$ *for all* $k < L$ *and* $(x, a, x') \in X_k \times A \times X_{k+1}$*, and*

$$\widehat{q}_{t+1} = \arg \min_{q \in \Delta(\mathcal{P}_t)} \widehat{\ell}_t^{\top} q + \frac{1}{\eta} D(q \| \widehat{q}_t),$$

*where* $D(q \| q') = \sum_{x,a,x'} q(x, a, x') \ln \frac{q(x,a,x')}{q'(x,a,x')} - \sum_{x,a,x'} (q(x, a, x') - q'(x, a, x'))$ *ensures*

$$\sum_{t=1}^{T} \widehat{\ell}_t^{\top} (\widehat{q}_t - q) \leq \frac{L \ln (|X|^2 |A|)}{\eta} + \eta \sum_{t,x,a} \widehat{q}_t(x, a) \widehat{\ell}_t(x, a)^2,$$

*for any* $q \in \cap_t \Delta(\mathcal{P}_t)$*, as long as* $\widehat{\ell}_t(x, a) \geq 0$ *for all* $t, x, a$*.*

