# OpenReview forum: "Learning Adversarial MDPs with Stochastic Hard Constraints"
_ICML.cc/2025/Conference — ICML 2025 poster_

### Official Review · Reviewer_RDix · 2025-02-25

**Overall Recommendation:** 3

**Summary:**

This paper considers adversarial MDP problems with stochastic constraints. The paper seeks to bound both the regret and the violations (hard). The paper shows that $O(\sqrt{T})$ regret and $O(\sqrt{T})$ violations bound only when the Slater's condition is satisfied. The paper shows that $O(\sqrt{T})$ regret and no violation when a strictly feasible policy is known. In fact, it also shows that it is possible to achieve constant violation and $O(\sqrt{T})$ regret only when one does not know the strict feasibility parameter.

**Claims And Evidence:**

The paper is theoretical in nature. All the claims have been proved.

**Essential References Not Discussed:**

The paper has not discussed with some of the major papers in the CMDP domain. While I agree that those papers did not consider adversarial rewards, however, in order to understand the technical contributions, one needs to understand the technical novelties required. In particular, why not combining all these approaches will not be enough to solve this problem?

[A1]. Efroni, Yonathan, Shie Mannor, and Matteo Pirotta. "Exploration-exploitation in constrained mdps." arXiv preprint arXiv:2003.02189 (2020).

[A2]. Bura, A., HasanzadeZonuzy, A., Kalathil, D., Shakkottai, S. and Chamberland, J.F., 2022. DOPE: Doubly optimistic and pessimistic exploration for safe reinforcement learning. Advances in neural information processing systems, 35, pp.1047-1059.

[A3]. Liu, T., Zhou, R., Kalathil, D., Kumar, P. and Tian, C., 2021. Learning policies with zero or bounded constraint violation for constrained mdps. Advances in Neural Information Processing Systems, 34, pp.17183-17193.

[A4]. Ghosh, Arnob, Xingyu Zhou, and Ness Shroff. "Towards achieving sub-linear regret and hard constraint violation in model-free rl." In International Conference on Artificial Intelligence and Statistics, pp. 1054-1062. PMLR, 2024.

**Experimental Designs Or Analyses:**

Not applicable.

**Methods And Evaluation Criteria:**

Not applicable as the paper's scopes are theoretical in nature.

**Other Comments Or Suggestions:**

Please see the questions

**Other Strengths And Weaknesses:**

Strengths:

1. The paper is well written.

2. The proof ideas seem to be correct.

Weaknesses:

1. The major weakness is that the technical novelties are not clear. CMDP setup is well studied. As I discussed, it is not clear what are the major technical innovations.

2. The paper only considers tabular case.

**Post Rebuttal:**

This work is solid and I am convinced about the contributions. I have raised my score.

**Questions For Authors:**

1. In the guaranteeing safety Algorithm (S-OPS Section 5), the paper utilizes a random exploration with a certain probability in the first episode, and also when the state-action occupancy found by the algorithm is not safe. However, is there any assumption which states that random exploration is safe? In general, such an exploration might not be safe.

2. The algorithm relies on building optimistic state-action-occupancy measures, how easy (computationally) it is?

3. What are the values of $\lambda_t$? How they are refined with time $t$?

**Relation To Broader Scientific Literature:**

This paper seeks to contribute in the space of adversarial MDP with stochastic constraints. The paper extends the adversarial MDP work to the CMDP setup with stochastic constraints. The paper achieves optimal regret and violation bound.

**Theoretical Claims:**

I briefly checked the correctness which seems to be correct.

---

> ### Author Rebuttal · Authors · 2025-03-31
>
> We thank the Reviewer for the effort in evaluating our work.
>
> > On the comparison with existing works and [W1]
>
> [A1] focuses on CMDPs with stochastic rewards and constraints, showing how it is possible to achieve $\sqrt T$ regret and violation. The only technique proposed by [A1] which is of interest in our setting is the optimistic safe set estimation, which is employed by SV-OPS.
>
> [A2] and [A3] study the stochastic version of our second scenario. Their randomization techniques cannot work in our second scenario, since in adversarial settings, it is not possible to play the safe policy for a fixed amount of rounds and then switch to an adversarial online learning algorithm on a pessimistic decision space. To see this, notice that pessimistic decision spaces increase over time and that adversarial online learning algorithms do not work on increasing decision spaces in $t$.
>
> Moreover, notice that even if [A3] shows how to obtain constant violation, but this is done employing a weaker version of constraints violation (which allows for cancellations between episodes), thus they are not applicable to our third scenario.
>
> Similarly to [A1], [A4] focuses on CMDPs with stochastic rewards and constraints, showing how it is possible to achieve $\sqrt T$ positive regret and violation. Their techniques cannot be generalized to adversarial settings, since they assume that there exists an underlying reward distribution in their analysis. Moreover, notice that the algorithm proposed in [A4] has an exponential worst-case running time, while our algorithm is polynomial.
>
> To conclude, our paper's main contributions are highlighted as follows. First, we study a novel CMDP setting, encompassing both adversarial losses and hard constraints. Specifically, we focus on three scenarios, which are all novel and cannot be tackled by existing algorithms. The first one is novel in terms of results, while we agree that we combine existing techniques to get our theoretical results. The second and third scenarios are instead novel both in terms of results and in terms of algorithmic techniques. Furthermore, the third scenario has never been studied even in simpler stochastic settings. Finally, we propose a novel lower bound which can be of independent interest, since it applies to the stochastic setting, too.
>
> We will surely include this discussion in the final version of the paper.
>
> > [W2]
>
> We agree with the Reviewer that studying CMDPs with infinite state-action spaces is an interesting future direction. Nevertheless, since this is the first work to tackle both adversarial losses and hard constraints, providing a large set of different results, we believe it is still of interest for the community.
>
> > [Q1]
>
> As is standard in the adversarial MDPs literature, the performance metrics are computed taking the expectation over policies and transitions. Since S-OPS plays non-Markovian policies, the expected violation is computed taking into account the randomization selected by the algorithm (please refer to Theorem 5.1. for additional details).
>
> > [Q2]
>
> Optimistic occupancy measures can be computed in $O(|X|^3|A|^2)$ steps (please refer to Jin et al. (2020) for the associated algorithm pseudocode).
>
> > [Q3]
>
> $\lambda_t$ is proportional to the uncertainty the algorithm has on both constraints and transitions. Thus, $\lambda_t$ is greater in the first episodes while converges to $0$ as the learning dynamic evolves.
>
> Since these are crucial aspects of our work, please let us know if further details are necessary. We would be happy to engage in further discussion.

---

> > ### Comment · Reviewer_RDix · 2025-04-03
> >
> > I would like to thank the authors for their responses. Now, the contributions become much more clearer. I have gone over the paper again, and have a few more comments which are required to understand the paper in a clearer way.
> >
> > 1. My first comment related to the second setting that you consider, i.e., ensuring that safety is not violated in the exploration. The basic idea is similar to the DOPE paper [A1] If you see the DOPE work where one uses a mixture policy. The mixture is based on the pessimistic estimate and the safe policy. The key is that because of the adversarial loss, you have to update the state-action-occupancy measure in a different way, that contribution is clear to me from the first setting now.
> >
> > Now, coming to the question--> The DOPE paper has to use double optimism in the reward (an additional bonus term) because of the pessimism used in the state-action occupancy measure. However, here, you are not using it to achieve a sub-linear. What is the reason behind it?
> >
> > 2. Also, regarding the third setting, it is very difficult to get any intuition. Can you please explain how you are estimating the safe policy and the value function corresponding to it in a constant number of episodes? Once you get a safe policy, you can apply the algorithm in the second setting, hence, understanding the question is very important.
> >
> > [A1]. Bura, A., HasanzadeZonuzy, A., Kalathil, D., Shakkottai, S. and Chamberland, J.F., 2022. DOPE: Doubly optimistic and pessimistic exploration for safe reinforcement learning. Advances in neural information processing systems, 35, pp.1047-1059.

---

> > > ### Author Response · Authors · 2025-04-03
> > >
> > > > The DOPE paper has to use double optimism in the reward (an additional bonus term) because of the pessimism used in the state-action occupancy measure. However, here, you are not using it to achieve a sub-linear. What is the reason behind it?
> > >
> > > We thank the Reviewer for the opportunity to better clarify this aspect. While our approach and the one of [A1] may seem similar since both of them employ strategy mixtures to guarantee the safety property, there are more than one key differences.
> > >
> > > Specifically, the main idea of Dope is the following. The strictly safe policy is played for a certain amount of time, after which a good estimate on constraints costs and transitions is available. This allows the pessimistic set to be large enough to use it. Indeed, notice that the pessimistic safe set is empty in the first episodes, when no information about the environment is available. This approach requires optimism on reward and transition, while pessimism on the costs, as pointed out by the Reviewer, to properly tackle the exploration-exploitation trade off.
> > > To summarize, all the techniques employed in [A1] are required because they directly optimize over the pessimistic decision space (see (10) in their paper).
> > >
> > > Optimizing over the pessimistic decision space cannot be done in our setting, since adversarial no-regret algorithms do not work in increasing decision spaces. Thus, our idea is pretty different. We do not play the strictly safe policy for a fixed amount of time. Instead, at each episode, we first allow the algorithm to select a policy in an optimistic manner, that is, employing the optimistic safe set as the decision space of the OMD update. Notice that, the policy selected optimistically is clearly no-regret given the results shown for SV-OPS. Then, we combine this policy with the strictly safe one, selecting the combination factor such that the final output policy is safe with high probability while being as close as possible to the one obtained with the OMD update. Finally, showing that the combination factor is lower-bounded by a constant factor allows us to get the final result of sublinear regret.
> > >
> > > We hope the explanation above addressed the Reviewer concern.
> > >
> > > > Also, regarding the third setting, it is very difficult to get any intuition. Can you please explain how you are estimating the safe policy and the value function corresponding to it in a constant number of episodes?
> > >
> > > We thank the Reviewer for the interesting question. Indeed, the fundamental reason why our approach works is that our technique only needs to estimate the safe policy within a constant **multiplicative** factor. Our result may look surprising since, most of the time, we focus on minimizing additive regret. Here, instead, we aim at the far less challenging goal of finding a strategy that is a constant fraction approximation of the optimal one (in our case, the strictly safe one). Intuitively, after a constant number of rounds $\tau$, the regret with respect to the most feasible policy is of order $\sqrt{\tau}$, which is only a constant fraction of $\tau \rho$. Hence, the per round feasibility margin is $\frac{\tau \rho- \sqrt{\tau}}{\tau}= \Omega(\rho)$.
> > >
> > > We will surely include this discussion in the final version of the paper and we hope to have properly addressed the Reviewer concern.

---

### Official Review · Reviewer_Lzh6 · 2025-03-04

**Overall Recommendation:** 3

**Summary:**

This paper studies online learning in constrained Markov Decision Processes with adversarial losses and stochastic hard constraints under bandit feedback. The authors introduce novel algorithms for three distinct scenarios of CMDPs, ensuring sublinear regret while managing constraint violations in different ways. The authors provide theoretical guarantees for the proposed algorithms.

**Claims And Evidence:**

The core theoretical claims (sublinear regret, constraint violation guarantees, and the lower bound) are well supported by clear mathematical proofs.

**Essential References Not Discussed:**

I did not find any essential references missing.

**Experimental Designs Or Analyses:**

The paper does not include any experimental results.

**Methods And Evaluation Criteria:**

The regret and constraint violation are standard evaluation criteria for online CMDPs.

**Other Comments Or Suggestions:**

I suggest adding a discussion on the computational complexity of the proposed algorithms and comparing them against alternative CMDP approaches. Moreover, a detailed comparison with existing methods and a clear exposition of the novel components in the algorithms would enhance the paper's clarity and impact.

**Other Strengths And Weaknesses:**

### Strengths
1. This paper is the first to study CMDPs with both adversarial losses and stochastic hard constraints, extending previous work on constrained RL.
2. This work presents different algorithms tailored to distinct CMDP scenarios and provides rigorous theoretical guarantees for each.
3. The paper establishes a lower bound, highlighting the fundamental trade-off between constraint satisfaction and regret minimization.

### Weaknesses
1. The paper lacks experimental results, making it difficult to assess the practical effectiveness and computational feasibility of the proposed algorithms.
2. The paper could benefit from a more detailed discussion of the computational complexity of the proposed algorithms, for example, the KL-divergence-based projection in SV-OPS and S-OPS (Equation 2).
3. The paper does not compare its work against alternative CMDP approaches, such as Lagrangian-based methods, which relax constraints using dual variables and model-based safe RL techniques that explicitly incorporate safety models. Moreover, a discussion on how existing methods fail to handle the hard constraints would help justify the novelty of the proposed approach.
4. Distinguishing the new components from existing techniques in the algorithms would help clarify the novelty of the proposed methods.  For example, the upper occupancy bound, optimistic loss estimator, and OMD in SV-OPS appear similar to Jin et al. (2020) [1], which should be explicitly highlighted. Since I am not deeply familiar with the CMDP literature, it is unclear whether the constraint-handling components are new or adaptations of prior techniques.

[1]. Jin et al., Learning adversarial Markov decision processes with bandit feedback and unknown transition. In ICML 2020.

**Questions For Authors:**

NA

**Relation To Broader Scientific Literature:**

The paper is well-positioned in the literature on online CMDPs.

**Theoretical Claims:**

The theoretical claims are well-supported by the proofs, though I do not have time to check the details of the proofs.

---

> ### Author Rebuttal · Authors · 2025-03-31
>
> We thank the Reviewer for the effort in evaluating our work.
>
> >  W1
>
> We agree with the Reviewer that experiments are always beneficial; nevertheless, we underline that both in the online CMDPs literature and the adversarial MDPs one, many works do not have experimental results (see e.g., Rosenberg et al. (2019b), Jin et al. (2020), Efroni et al. (2020), Stradi et al. (2024b) and many others). Despite the lack of experimental evaluation, many of the aforementioned works (and many others) have been published in top AI conferences, such as ICML.
>
> > W2
>
> We thank the Reviewer for the opportunity to clarify this aspect. Please notice that Eq. (2) is a convex program with linear constraints, which can be approximated arbitrarily well in polynomial time. We underline that solving this kind of projection is standard in the adversarial MDPs literature (see e.g., Rosenberg et al. (2019b), Jin et al. (2020)). Similarly, it is standard for adversarial online learning algorithms to require projections to be solved at each episode, even in easier single-state full-feedback settings (e.g., OGD and OMD algorithms).
>
> > W3
>
> We thank the Reviewer for the opportunity to clarify these aspects.
>
> Regarding Lagrangian methods, state-of-the-art primal-dual algorithms are not suited to handle any of our scenarios. Broadly, these algorithms can be categorized into two main families: (i) primal-dual methods designed specifically for stochastic settings (e.g., [1], [2], [3]), and (ii) algorithms that address both stochastic and adversarial CMDPs (e.g., [4], [5]). All the algorithms in the first class assume that the losses/rewards are stochastic in nature, thus, their techniques and results are not applicable to our three scenarios. For the latter, [4], [5] achieve $\sqrt T$ regret and violation when the constraints are stochastic and the losses adversarial. Nevertheless, notice that their violation definition allows per-episode violations to cancel out, thus they employ a weaker violation definition w.r.t. ours. Moreover, the aforementioned papers assume Slater’s condition to hold, which is not the case for our first scenario. Our second and third scenarios have never been tackled employing primal-dual methods even in simpler stochastic rewards settings.
>
> As concerns model-based techniques, our work is the first to tackle adversarial losses. Additionally, our work is the first one to provide constant violation bounds (employing our positive violation definition) without assuming the knowledge of a safe policy, even considering stochastic settings. Finally, existing model-based algorithms tailored for stochastic versions of our second scenario do not work in adversarial settings, and cannot be easily extended due to a different randomization approach ([6], [7]). For further details on these aspects please refer to the next answer.
>
> We will surely include this discussion in the final version of the paper.
>
>
> > W4
>
> We thank the Reviewer for the opportunity to better discuss the algorithmic components of our techniques.
>
> SV-OPS employs the optimization approach of Jin et al. (2020) on optimistic safe decision spaces. Optimistic safe decision spaces were originally introduced by [1] in the context of stochastic CMDPs.
> The idea behind S-OPS and its randomization approach is novel in the CMDPs literature. We acknowledge that there exist other forms of randomization which are suitable for stochastic CMDPs (see, e.g., [6], [7]), but they fail in working in adversarial settings. To see this, notice that the algorithms presented in [7], [8] play the safe policy for a fixed amount of rounds and then resort to pessimistic decision spaces which are increasing in $t$. It is well-known that adversarial online learning algorithms do not work on increasing decision spaces. Thus, their approach cannot be easily generalized to tackle adversarial loss.
> BV-OPS employs techniques which are completely novel in the literature.
>
> We hope to have properly addressed the Reviewer concerns. Please let us know if further discussion is necessary.
>
>
> [1] Efroni et al (2020), “Exploration-Exploitation in Constrained MDPs”
>
> [2] Müller et al. (2024), “Truly No-Regret Learning in Constrained MDPs”
>
> [3] Stradi et al. (2024), “Optimal Strong Regret and Violation in Constrained MDPs via Policy Optimization”
>
> [4] Qiu et al. (2020), “Upper confidence primal-dual reinforcement learning for cmdp with adversarial loss.”
>
> [5] Stradi et al. (2024), “Online Learning in CMDPs: Handling Stochastic and Adversarial Constraints”
>
> [6] Liu et al. (2021), “Learning policies with zero or bounded constraint violation for constrained mdps”
>
> [7] Bura et al. (2022), “DOPE: Doubly optimistic and pessimistic exploration for safe reinforcement learning.”

---

### Official Review · Reviewer_jA7c · 2025-03-10

**Overall Recommendation:** 4

**Summary:**

This paper studies episodic Constrained Markov Decision Processes (CMDPs) with adversarial losses and stochastic hard constraints under bandit feedback. It is the first to address a setting that combines both adversarial losses and strict hard constraints, whereas prior work has either considered adversarial losses with soft constraints, allowing negative violations to cancel out with positive ones, or stochastic losses with hard constraints. The authors propose three algorithms tailored to different scenarios and under different assumptions: the first ensures sublinear regret while keeping cumulative constraint violations sublinear, the second guarantees that the constraints are satisfied at every episode assuming the learner knows a strictly feasible policy, and the third achieves constant violation regret assuming a strictly feasible policy exists but is not known to the learner.

**Claims And Evidence:**

All theoretical claims are followed by proofs in the appendix.

**Essential References Not Discussed:**

Related work is clear.

**Experimental Designs Or Analyses:**

not applicable.

**Methods And Evaluation Criteria:**

As a theoretical paper, the algorithms make sense for the problem.

**Other Comments Or Suggestions:**

No further comments

**Other Strengths And Weaknesses:**

- **Strenghts**: First, the paper is well-written. Second, the extension of SV-OPS to scenarios that ensure safety at evert episode and guarantee constant violation appears novel and may be of significant interest to the online C-MDP community.

 - **Weaknesses**: For SV-OPS, the sublinear constraint violation is ensured due to the accurate estimation of $ G_t $ and the projection in Eq. (2). However, implementing this in practice would require access to a solver for a linear programming problem with at least $ \Omega(XLA) $ decision variables and constraints, making it computationally expensive for frameworks with large state spaces.

**Questions For Authors:**

Regarding the concern raised above about the projection, could it be feasible to design a primal-dual algorithm (which performs well in practice) that employs a similar OMD iterative scheme to handle adversarial losses while achieving both sublinear regret and sublinear violations?

**Relation To Broader Scientific Literature:**

This paper is the first to tackle the case of constrained MDPs when facing both the challenges of adversarial losses and hard constraints. Existing results so far would either treat adversarial losses with soft constraints, or stochastic losses with hard constraints.

However, the SV-OPS algorithm from Section 4 (the first algorithm proposed) primarily builds on existing ideas from the online MDP literature, introducing only minor generalizations. The algorithm follows the same OMD iterative scheme as in [Jin 2020]. But instead of projecting onto the space of occupancy measures that satisfy the dynamic constraints for any probability transition within a confidence set around the true transition, SV-OPS also projects onto the set of estimated constraints to ensure sublinear positive constraints violation. Projecting on the estimated constraints has already been introduced by [Efroni et al. 2020] in the set of stochastic losses.

On the other hand, the algorithm proposed in Section 5, which ensures safety in every episode, and the one proposed in Section 6, which guarantees constant violations, introduce novel ideas that may be of interest to the broader online C-MDP community.

**Theoretical Claims:**

Proofs for theoretical claims seem correct.

---

> ### Author Rebuttal · Authors · 2025-03-31
>
> We thank the Reviewer for the positive evaluation and for the interesting question. Indeed, this is an interesting future direction, nonetheless, we believe that it would be highly non-trivial to employ a primal-dual approach to solve any of the scenarios we study, due to the following reasons. For the first scenario, primal-dual methods generally require Slater’s condition to hold, which is not the case of our algorithm. Moreover, primal-dual methods generally fail to remove the $1/\rho$ dependence in the regret and violation bound, thus achieving worse regret and violation guarantee than SV-OPS. Finally, notice that primal-dual methods developed for adversarial loss settings do not guarantee sublinear positive violation, but employ a violation definition where the cancellations are allowed (see, e.g., Stradi et al. (2024b)). As concerns the second and third scenario, we believe that the randomization with the strictly feasible policy requires an optimistic safe-set estimation, which is generally not employed in primal-dual methods.
>
> Finally we want to underline that solving this kind of projection is standard in the adversarial MDPs literature (see e.g., Rosenberg et al. (2019b), Jin et al. (2020)). Similarly, it is standard for adversarial online learning algorithms to require projections to be solved at each episode, even in easier single-state full-feedback settings (e.g., OGD and OMD algorithms).
>
> Please let us know if further discussion is necessary.

---

### Official Review · Reviewer_39LP · 2025-03-11

**Overall Recommendation:** 3

**Summary:**

This paper introduces algorithms for constrained Markov Decision Processes (MDPs) with stochastic hard constraints, considering different assumptions and objectives for constraint violations. Specifically, it examines three key cases: (1) when constraints are feasible, (2) when constraints are strictly feasible with a known feasible policy, and (3) when constraints are strictly feasible but no strictly feasible policy is known beforehand. The paper establishes sublinear regret and constraint violation bounds for all three scenarios. The writing is clear, and the proofs are straightforward to follow. The core approach involves optimistically satisfying cost constraints in the first case and randomizing between an optimistic and a strictly feasible policy in the second.

**Claims And Evidence:**

The lower bound in Theorem 6.6 seems a bit puzzling to me. Please refer to the questions for the authors field below.

**Essential References Not Discussed:**

[1] Sinha, Abhishek, and Rahul Vaze. "Optimal algorithms for online convex optimization with adversarial constraints." Advances in Neural Information Processing Systems 37 (2025): 41274-41302.

**Experimental Designs Or Analyses:**

No experimental result has been provided.

**Methods And Evaluation Criteria:**

This is fine.

**Other Comments Or Suggestions:**

Statements on Constraint Violation and Regret appear first in Theorems 4.2 and 4.3, but I could not find their formal definitions earlier in the paper. Clearly stating these definitions upfront would improve readability and help the reader follow the theoretical results more easily.

**Other Strengths And Weaknesses:**

N/A

**Questions For Authors:**

1.	Prior work [1] demonstrated that $O(\sqrt{T})$ regret and violation bounds can be achieved in online convex optimization with adversarial hard constraints without assuming Slater’s condition (i.e., when $\rho=0$). Keeping this result in view, could the authors offer some intuition on why Slater’s condition assumption is necessary for online reinforcement learning with hard constraints? Specifically, why does the result in [1] not contradict the lower bound given in Theorem 6.6, given that the lower bound proof involves a single state MDP?

2.	The paper does not include any numerical comparisons with prior algorithms in the MDP setting, making it difficult to assess the practical effectiveness of the proposed theoretical results.

3.	In Equation (2), computing the projection may be computationally challenging. Can the authors provide an upper bound on its worst-case computational complexity?

References:

[1] Sinha, Abhishek, and Rahul Vaze. "Optimal algorithms for online convex optimization with adversarial constraints." Advances in Neural Information Processing Systems 37 (2025): 41274-41302.

**Relation To Broader Scientific Literature:**

Please refer to the questions for the authors field below.

**Theoretical Claims:**

This is mostly fine.

---

> ### Author Rebuttal · Authors · 2025-03-31
>
> We thank the Reviewer for the positive evaluation of the paper.
>
> > Question 1.
>
> We thank the Reviewer for the opportunity to clarify this aspect. Indeed, there is no contradiction between [1] and our lower bound, since our lower bound holds for our second and third scenario only, namely, when we aim to guarantee a violation of order $o(\sqrt{T})$, such as $0$ or constant violation. In our first scenario, we **do not assume Slater's condition** and we guarantee $O(\sqrt{T})$ regret and violation, thus, there is no contradiction w.r.t. [1].
>
> > Question 2.
>
> We agree with the Reviewer that experiments are always beneficial; nevertheless, we underline that both in the online CMDPs literature and the adversarial MDPs one, many works do not have experimental results (see e.g., Rosenberg et al. (2019b), Jin et al. (2020), Efroni et al. (2020), Stradi et al. (2024b) and many others). Despite the lack of experimental evaluation, many of the aforementioned works (and many others) have been published in top AI conferences, such as ICML.
>
> > Question 3.
>
> We thank the Reviewer for the opportunity to clarify this aspect. Please notice that Eq. (2) is a convex program with linear constraints, which can be approximated arbitrarily well in polynomial time. We underline that solving this kind of projection is standard in the adversarial MDPs literature (see e.g., Rosenberg et al. (2019b), Jin et al. (2020)). Similarly, it is standard for adversarial online learning algorithms to require projections to be solved at each episode, even in easier single-state full-feedback settings (e.g., OGD and OMD algorithms).
>
> We finally thank the Reviewer for the other comments and suggestions, we will surely update the final version of the paper taking them into account.

---

### Decision · Program_Chairs · 2025-05-01

**Decision:**

Accept (poster)

**Comment:**

We thank the authors for their submission.
The reviews appreciated the algorithmic techniques and the end results.
There are some issues mentioned in the reviews that should be clarified in the next version, for example:
- Detailed description of the novelty compared to prior work.
- Discussion of computational complexity of the proposed algorithms.
- Discussion on why existing methods fail under hard constraints.